

# Parameter sensitivity analysis of dynamic ice sheet models-Numerical computations

Gong Cheng[1] and Per Lötstedt[1]

[1]Department of Information Technology, Uppsala University, P. O. Box 337, SE-75105 Uppsala, Sweden

**Correspondence:** Gong Cheng (cheng.gong@it.uu.se)

**Abstract.** The friction coefficient and the base topography of a stationary and a dynamic ice sheet are perturbed in two models for the ice: the full Stokes equations and the shallow shelf approximation. The sensitivity to the perturbations of the velocity and the height at the surface is quantified by solving the adjoint equations of the stress and the height equations providing weights for the perturbed data. The adjoint equations are solved numerically and the sensitivity is computed in several examples in two dimensions. Comparisons are made with analytical solutions to simplified problems.

## 1 Introduction

The result of isothermal simulations of large ice sheets depends on the ice model, the topography, and the parametrization of the conditions at the base of the ice. The models are systems of partial differential equations (PDEs) for the velocity, pressure, and height of the ice. The topography and the friction model with its parameters determine the horizontal velocity and the height at the ice surface in the computations. In the inverse problem, the parameters at the base are inferred from data at the surface by solving adjoint equations and minimizing the difference between given data and simulated results. In this paper, we estimate the sensitivity of the surface observations to changes in the basal conditions by solving the adjoint equations to the full Stokes (FS) equations and the shallow shelf (or shelfy stream) approximation (SSA), see Greve and Blatter (2009); MacAyeal (1989). The advantage of solving the adjoint equations in a variational *control* method is that the effect of many perturbations of the parameters at the bottom is obtained for one observation at one point of the surface at a certain time point. If there are many observations and only one perturbation, then it is more efficient to compute the sensitivity by solving the forward model PDEs twice in a *direct* method, firstly with the unperturbed parameters, secondly with the perturbed parameters, and then take the difference between the solutions. The direct method has the advantage that there is no need to implement a solver for the adjoint equations.

Most methods for inversion of ice surface data to compute parameters in the models at the ice base rely on a solution of the adjoint stress equation with a given fixed geometry of the ice as in MacAyeal (1993); Petra et al. (2012). The time dependent height equation for the moving upper surface is not included in the inversion. The stationary basal friction coefficients have been derived from satellite data in this way for many glaciers and continental ice sheets using velocity data in e.g. Gillet-Chaulet et al. (2016); Isaac et al. (2015); Schannwell et al. (2019); Sergienko and Hindmarsh (2013). The sensitivity to changes at the base increases closer to the grounding line in the coastal regions in Durand et al. (2011). The base topography is inferred from




height data in van Pelt et al. (2013) without solving the adjoint equations. The conditions between the ice and the bedrock vary in time and sometimes the friction parameter varies several orders of magnitude in a decade in Jay-Allemand et al. (2011). In addition, there are variations on seasonal and diurnal time scales with examples in Schoof (2010); Shannon et al. (2013); Vallot et al. (2017). Other time dependent forces are considered in Seddik et al. (2019). The effect of a seasonal variation of

the lubrication at the base of the ice is studied in Shannon et al. (2013) for the Greenland ice sheet by solving the FS and other high order equations. Fast temporal variations in the meltwater under the ice drive the ice flow in the analysis in Schoof (2010). The spatial and temporal variations of the basal conditions are inferred from satellite data in Larour et al. (2014) with an inverse method for SSA and automatic differentiation. Based on observations, the conclusion in Sole et al. (2011) is also that the annual change of the water drainage under the ice affects the sliding and the acceleration and deceleration of the ice. Here,

we solve the adjoint equations to both the stress equation and the time dependent height equation in FS and SSA to examine how the dynamics of the models change the sensitivity to the base parameters. The adjoint equations are derived and analytical solutions are found to simplified equations in a companion paper by Cheng and Lötstedt (2019).

The forward advection equation for the height and the stress equations for the velocity for FS are here solved numerically in two dimensions (2D) with Elmer/Ice (Gagliardini et al. (2013); Gillet-Chaulet et al. (2012)). The solver of the adjoint stress

equation in Elmer/Ice is amended by the adjoint height equation. The forward and adjoint SSA equations are solved in 2D by a finite difference method. The perturbations are observed in the velocity and the height at certain points in space and time. Comparisons are made for steady state and time dependent problems between a direct calculation of the change at the ice surface and using the control technique with the adjoint solution. Simplified adjoint stress equations have been proposed and used in Martin and Monnier (2014); Morlighem et al. (2013); Mosbeux et al. (2016). The sensitivity in the SSA model

is evaluated here for such simplifications in the adjoint SSA equations. The numerical solutions are also compared to the analytical formulas in Cheng and Lötstedt (2019). There is a transfer matrix between the perturbations in the parameters at the base and the observations at the surface. The properties of this matrix are evaluated to see which combinations of perturbations and observations that are well and ill-conditioned. In an ill-conditioned problem, the sensitivity is low at the surface to perturbations at the base. This matrix can be used to quantify the uncertainty in the ice flow due to uncertainties in

the model parameters, see e.g. Bulthuis et al. (2019); Schlegel et al. (2018); Smith (2014).

The ice equations and the corresponding adjoint equations for FS and SSA are given in Sect. 2. The computed sensitivities are compared for the direct method and the control method in Sect. 3 for steady state and time dependent problems in 2D. The ice configuration is taken from the MISMIP benchmark project in Pattyn et al. (2012). The results are discussed and conclusions are drawn in Sections 4 and 5. Formulas from Cheng and Lötstedt (2019) are found in Appendix A.

Vectors and matrices are written in bold as $\mathbf{a}$ and $\mathbf{A}$. The operations $\otimes, :,$ and $\star$ on vectors $\mathbf{a}$ and $\mathbf{c}$, matrices $\mathbf{A}$ and $\mathbf{C}$, and four index tensors $\mathcal{A}$ are defined by

$$(\mathbf{a} \otimes \mathbf{c})_{ij} = a_i c_j, \quad \mathbf{a} : \mathbf{c} = \mathbf{a} \cdot \mathbf{c} = \sum_i a_i c_i,$$
$$(\mathbf{A} \otimes \mathbf{C})_{ijkl} = A_{ij} C_{kl}, \quad \mathbf{A} : \mathbf{C} = \sum_{ij} A_{ij} C_{ij}, \quad (\mathcal{A} \star \mathbf{C})_{ij} = \sum_{kl} \mathcal{A}_{ijkl} C_{kl}. \tag{1}$$

The norm of a vector $\mathbf{a}$ is defined by $\|\mathbf{a}\| = (\mathbf{a} \cdot \mathbf{a})^{1/2}$.





## 2 Ice models

The equations of two ice models and their adjoint equations are stated in this section. The FS equations are considered to be an accurate model of ice sheets and the SSA equations are an approximation of the FS equations suitable e.g. for fast flowing ice on the ground and ice floating on water, see Greve and Blatter (2009).

### 2.1 Full Stokes equations

The FS equations are a system of PDEs for the velocity of the ice $\mathbf{u}(\mathbf{x}, t) = (u_1, u_2, u_3)^T$, the pressure $p(\mathbf{x}, t)$, and the height $h(x, y, t)$ with the coordinates $\mathbf{x} = (x, y, z)$ and time $t$. There is a stress equation satisfied by $\mathbf{u}$ and $p$ and an advection equation for $h$. The adjoint equation of the stress equation is derived in Petra et al. (2012) and the adjoint equations of the stress and the height equations are found in Cheng and Lötstedt (2019). The sensitivity of observations of the velocity and the height of the ice surface is derived for perturbations in the friction coefficient at the ice base.

The domain of the ice is $\Omega$ with boundary $\Gamma$ in three dimensions (3D). The boundary consists of the ice surface at the upper boundary $\Gamma_s$, the lower boundary at the ice base $\Gamma_b$ and $\Gamma_w$, and the vertical, lateral boundaries $\Gamma_u$ and $\Gamma_d$ where $\Gamma_u$ is the upstream boundary with $\mathbf{n} \cdot \mathbf{u} \leq 0$ and $\Gamma_d$ is the downstream boundary with $\mathbf{n} \cdot \mathbf{u} > 0$. The normal of $\Gamma$ pointing outward is denoted by $\mathbf{n}$. The projection of $\Gamma_s$ and $\Gamma_b$ on the horizontal $x-y$ plane is $\omega$ and the projections of $\Gamma_u$ and $\Gamma_d$ are $\gamma_u$ and $\gamma_d$,

respectively. The $z$ coordinate of the grounded base $\Gamma_b$ is the topography and the bathymetry $b(x, y)$. The grounding line $\gamma_{GL}$ separates $\Gamma_b$ on $\omega$ from $\Gamma_w$ floating on water with a moving $z$-coordinate $z_b(x, y, t)$. Formal definitions of these domains are

$$
\begin{aligned}
\Omega &= \{\mathbf{x} | (x, y) \in \omega, b(x, y) \leq z \leq h(x, y, t)\}, \\
\Gamma_s &= \{\mathbf{x} | (x, y) \in \omega, z = h(x, y, t)\}, \\
\Gamma_b &= \{\mathbf{x} | (x, y) \in \omega, z = b(x, y), x < x_{GL}(y)\}, \\
\Gamma_w &= \{\mathbf{x} | (x, y) \in \omega, z = z_b(x, y, t), x > x_{GL}(y)\}, \\
\Gamma_u &= \{\mathbf{x} | (x, y) \in \gamma_u, b(x, y) \leq z \leq h(x, y, t)\}, \\
\Gamma_d &= \{\mathbf{x} | (x, y) \in \gamma_d, b(x, y) \leq z \leq h(x, y, t)\}.
\end{aligned}
\tag{2}
$$

Let $\mathbf{I}$ be the identity matrix. The projection of a vector on the tangential plane of $\Gamma_b$ is denoted by $\mathbf{T} = \mathbf{I} - \mathbf{n} \otimes \mathbf{n}$ as in Petra et al. (2012). In 2D, $\mathbf{x} = (x, z)^T$, $\omega = [0, L]$, $\gamma_u = 0$, and $\gamma_d = L$.

### 2.1.1 Forward equations

The definitions of the strain rate $\mathbf{D}$ and the viscosity $\eta$ of the ice are

$$
\mathbf{D} = \tfrac{1}{2}(\nabla \mathbf{u} + \nabla \mathbf{u}^T), \ \eta(\mathbf{u}) = \tfrac{1}{2} A^{-\frac{1}{n}} (\mathrm{tr} \mathbf{D}^2(\mathbf{u}))^\nu, \ \nu = \tfrac{1-n}{2n}.
\tag{3}
$$

The trace of $\mathbf{D}^2$ is $\mathrm{tr} \mathbf{D}^2$ and the rate factor $A$ depends on the temperature of the ice, here assumed to be constant in isothermal flow. The material constant $n > 0$ is given in Glen's flow law. Then the stress tensor is

$$
\boldsymbol{\sigma}(\mathbf{u}, p) = 2\eta \mathbf{D}(\mathbf{u}) - p\mathbf{I}.
\tag{4}
$$





Let $\rho$ be the density of the ice, $\mathbf{g}$ be the gravitational acceleration and $a$ be the accumulation/ablation rate on the surface $\Gamma_s$. The notation is simplified with the slope vectors $\mathbf{h} = (h_x, h_y, -1)^T$ in 3D and $\mathbf{h} = (h_x, -1)^T$ in 2D. A subscript $x, y, z,$ or $t$ on a variable denotes a partial derivative such that e.g. $h_x = \partial h/\partial x$. Then the forward FS equations for $h, \mathbf{u},$ and $p$ are

$$
\begin{aligned}
&h_t + \mathbf{h} \cdot \mathbf{u} = a, \ \text{ on } \Gamma_s, \\
&h(\mathbf{x}, 0) = h_0(\mathbf{x}), \ \mathbf{x} \in \omega, \quad h(\mathbf{x}, t) = h_\gamma(\mathbf{x}, t), \ \mathbf{x} \in \gamma_u, \\
&-\nabla \cdot \boldsymbol{\sigma}(\mathbf{u}, p) = -\nabla \cdot (2\eta(\mathbf{u})\mathbf{D}(\mathbf{u})) + \nabla p = \rho \mathbf{g}, \quad \nabla \cdot \mathbf{u} = 0, \text{ in } \Omega(t), \\
&\boldsymbol{\sigma} \mathbf{n} = \mathbf{0}, \ \text{ on } \Gamma_s, \\
&\mathbf{T} \boldsymbol{\sigma} \mathbf{n} = -C f(\mathbf{T} \mathbf{u}) \mathbf{T} \mathbf{u}, \quad \mathbf{n} \cdot \mathbf{u} = 0, \ \text{ on } \Gamma_b.
\end{aligned}
\tag{5}
$$

The initial data for $h$ are $h_0(\mathbf{x})$ and $h_\gamma(\mathbf{x}, t)$ is specified on the inflow boundary $\gamma_u$. The expression $C f(\mathbf{Tu})$ defines the friction law with variable coefficient $C(\mathbf{x}, t)$ and a function $f(\cdot)$ of the projected velocity $\mathbf{Tu}$, e.g. as in Weertman (1957) where

$$
f(\mathbf{u}) = \|\mathbf{u}\|^{m-1}, \quad m > 0.
\tag{6}
$$

The Dirichlet boundary conditions of $\mathbf{u}$ on $\Gamma_u$ and $\Gamma_d$ are set to be $\mathbf{u}_u$ and $\mathbf{u}_d$.

### 2.1.2   Adjoint equations

We observe a quantity

$$
\mathcal{F} = \int_0^T \int_{\Gamma_s} F(\mathbf{u}, h) \, \mathrm{d}\mathbf{x} \mathrm{d}t
\tag{7}
$$

at the surface $\Gamma_s$ when $t \in [0, T]$. For example, if the ice is in the steady state and $F(\mathbf{u}) = u_1 \delta(\mathbf{x} - \mathbf{x}_*)$ with the Dirac delta $\delta$ then the observation is the $x$ component of $\mathbf{u}$ at $\mathbf{x}_*$

$$
\mathcal{F} = \int_{\Gamma_s} F(\mathbf{u}) \, \mathrm{d}\mathbf{x} = u_1(\mathbf{x}_*).
$$

If $F(h) = h \delta(\mathbf{x} - \mathbf{x}_*)$ then the height is observed

$$
\mathcal{F} = \int_{\Gamma_s} F(h) \, \mathrm{d}\mathbf{x} = h(\mathbf{x}_*).
$$

The adjoint equations depend on the first variations $F_\mathbf{u}$ and $F_h$ of $F(\mathbf{u}, h)$ with respect to $\mathbf{u}$ and $h$. In the first example above, $F_\mathbf{u} = (\delta(\mathbf{x} - \mathbf{x}_*), 0, 0)^T$ and $F_h = 0$ and in the second example $F_\mathbf{u} = \mathbf{0}$ and $F_h = \delta(\mathbf{x} - \mathbf{x}_*)$.





The adjoint FS equations form a system of PDEs for the adjoint height $\psi$, the adjoint velocity $\mathbf{v}$, and the adjoint pressure $q$. There is an advection equation for $\psi$ and an adjoint stress equation for $\mathbf{v}$ and $q$ such that

$$
\begin{aligned}
&\psi_t + \nabla \cdot (\mathbf{u}\psi) - \mathbf{h} \cdot \mathbf{u}_z \psi = F_h + F_{\mathbf{u}} \cdot \mathbf{u}_z, \text{ on } \Gamma_s, \\
&\psi(\mathbf{x},T) = 0, \ \psi(\mathbf{x},t) = 0, \text{ on } \Gamma_d, \\
&-\nabla \cdot \tilde{\boldsymbol{\sigma}}(\mathbf{v},q) = -\nabla \cdot (2\tilde{\boldsymbol{\eta}}(\mathbf{u}) \star \mathbf{D}(\mathbf{v})) + \nabla q = \mathbf{0}, \quad \nabla \cdot \mathbf{v} = 0, \text{ in } \Omega(t), \\
&\tilde{\boldsymbol{\sigma}}(\mathbf{v},q)\mathbf{n} = -(F_{\mathbf{u}} + \psi\mathbf{h}), \text{ on } \Gamma_s, \\
&\mathbf{T}\tilde{\boldsymbol{\sigma}}(\mathbf{v},q)\mathbf{n} = -Cf(\mathbf{Tu})\left(\mathbf{I} + \mathbf{F}_b(\mathbf{Tu})\right)\mathbf{Tv}, \text{ on } \Gamma_b, \\
&\mathbf{n} \cdot \mathbf{v} = 0, \text{ on } \Gamma_b,
\end{aligned}
\tag{8}
$$

where the adjoint viscosity, adjoint stress, and linearized friction law in Eq. (8) are according to Petra et al. (2012)

$$
\begin{aligned}
\tilde{\boldsymbol{\eta}}(\mathbf{u}) &= \eta(\mathbf{u})\left(\mathcal{I} + \tfrac{1-n}{n\mathbf{D}(\mathbf{u}):\mathbf{D}(\mathbf{u})}\mathbf{D}(\mathbf{u}) \otimes \mathbf{D}(\mathbf{u})\right), \\
\tilde{\boldsymbol{\sigma}}(\mathbf{v},q) &= 2\tilde{\boldsymbol{\eta}}(\mathbf{u}) \star \mathbf{D}(\mathbf{v}) - q\mathbf{I}, \\
\mathbf{F}_b(\mathbf{Tu}) &= \tfrac{m-1}{\mathbf{Tu}\cdot\mathbf{Tu}}(\mathbf{Tu}) \otimes (\mathbf{Tu}).
\end{aligned}
\tag{9}
$$

The tensor $\mathcal{I}$ with four indices $ijkl$ is 1 when $i = j = k = l$ and 0 otherwise.

The perturbation of the observation in Eq. (7) with respect to a perturbation in the friction coefficient $C$ is

$$
\delta\mathcal{F} = \int_0^T \int_{\Gamma_b} f(\mathbf{Tu})\mathbf{Tu} \cdot \mathbf{Tv}\, \delta C\, \mathrm{d}\mathbf{x}\,\mathrm{d}t
\tag{10}
$$

involving the tangential projections of the forward and adjoint velocities $\mathbf{Tu}$ and $\mathbf{Tv}$ at the grounded ice base $\Gamma_b$. This expression is derived in Cheng and Lötstedt (2019) and Petra et al. (2012) via the perturbation of the Lagrangian of the system of equations and evaluating it at the forward and adjoint solutions.

Only perturbations in $C$ are considered here for the FS model. Via the Lagrangian, the result of perturbations $\delta b$ in the topography can be derived but the complexity of the adjoint Eq. (8) would increase considerably.

## 2.2 Shallow shelf approximation

In the shallow shelf approximation of the FS equations, the velocity is constant in the vertical direction and the pressure is given by the cryostatic approximation (Greve and Blatter (2009); MacAyeal (1989)). The sensitivity of observations of the velocity at the surface and the height to perturbations in friction coefficients and the base topography is quantified for the SSA model.

### 2.2.1 Forward equations

It is sufficient to solve for the horizontal velocity $\mathbf{u} = (u_1, u_2)^T$ when $\mathbf{x} = (x,y) \in \omega$ thus simplifying the 3D FS problem Eq. (5) considerably. The viscosity in the SSA is

$$
\eta(\mathbf{u}) = \frac{1}{2}A^{-\frac{1}{n}}\left(u_{1x}^2 + u_{2y}^2 + \frac{1}{4}(u_{1y} + u_{2x})^2 + u_{1x}u_{2y}\right)^{\nu} = \frac{1}{2}A^{-\frac{1}{n}}\left(\frac{1}{2}\mathbf{B}:\mathbf{D}\right)^{\nu},
\tag{11}
$$





where $\mathbf{B}(\mathbf{u}) = \mathbf{D}(\mathbf{u}) + \nabla \cdot \mathbf{u}\,\mathbf{I}$. The stress tensor $\varsigma(\mathbf{u})$ in SSA is defined by

$$\varsigma(\mathbf{u}) = 2H\eta\mathbf{B}(\mathbf{u}). \tag{12}$$

Let $\mathbf{n}$ be the outward normal vector of the boundary $\gamma$, $\mathbf{t}$ the tangential vector such that $\mathbf{n} \cdot \mathbf{t} = 0$, and $H = h - b$ the thickness of the ice. The friction law is defined as in the FS case in Eq. (6) where the basal velocity is replaced by the horizontal velocity

since the vertical variation is neglected in SSA. Under the floating ice shelf on $\Gamma_w$, $C = 0$ in the friction law.

The ice dynamics system is

$$
\begin{aligned}
&h_t + \nabla \cdot (\mathbf{u}H) = a, \ \ 0 \le t \le T, \ \mathbf{x} \in \omega, \\
&h(\mathbf{x}, 0) = h_0(\mathbf{x}), \ \mathbf{x} \in \omega, \quad h(\mathbf{x}, t) = h_\gamma(\mathbf{x}, t), \ \mathbf{x} \in \gamma_u, \\
&\nabla \cdot \varsigma - Cf(\mathbf{u})\mathbf{u} = \rho g H \nabla h, \ \mathbf{x} \in \omega, \\
&\mathbf{n} \cdot \mathbf{u}(\mathbf{x}, t) = u_{\mathrm{in}}(\mathbf{x}, t), \ \mathbf{x} \in \gamma_u, \quad \mathbf{n} \cdot \mathbf{u}(\mathbf{x}, t) = u_{\mathrm{out}}(\mathbf{x}, t), \ \mathbf{x} \in \gamma_d, \\
&\mathbf{t} \cdot \varsigma\mathbf{n} = -C_\gamma f_\gamma(\mathbf{t} \cdot \mathbf{u})\mathbf{t} \cdot \mathbf{u}, \ \mathbf{x} \in \gamma_g, \quad \mathbf{t} \cdot \varsigma\mathbf{n} = 0, \ \mathbf{x} \in \gamma_w,
\end{aligned}
\tag{13}
$$

where $u_{\mathrm{in}} \le 0$ and $u_{\mathrm{out}} > 0$ are the inflow and outflow normal velocities on $\gamma_u$ and $\gamma_d$ of the boundary $\gamma = \gamma_u \cup \gamma_d$. The friction on the lateral side of the ice $\gamma = \gamma_g \cup \gamma_w$ depends on the tangential velocity $\mathbf{t} \cdot \mathbf{u}$ there. The friction law $C_\gamma f_\gamma(\mathbf{t} \cdot \mathbf{u})$ on $\gamma_g$ is

10 not necessarily the same as $Cf(\mathbf{u})$ on $\omega$.

The structure of the SSA system Eq. (13) is similar to the FS equations in Eq. (5). However, the velocity $\mathbf{u}$ is not divergence free in SSA and $\mathbf{B} \ne \mathbf{D}$ due to the cryostatic approximation.

### 2.2.2 Adjoint equations

The adjoint SSA equations are derived in Cheng and Lötstedt (2019) as in Sect. 2.1.2 by forming the Lagrangian and partial

integration using the forward equations and the boundary conditions in Eq. (13). The adjoint viscosity $\tilde{\eta}$ and adjoint stress $\tilde{\varsigma}$ are defined by

$$
\begin{aligned}
\tilde{\boldsymbol{\eta}}(\mathbf{u}) \ &= \eta(\mathbf{u})\left(\mathcal{I} + \frac{1-n}{n\mathbf{B}(\mathbf{u}):\mathbf{D}(\mathbf{u})}\mathbf{B}(\mathbf{u}) \otimes \mathbf{D}(\mathbf{u})\right), \\
\tilde{\varsigma}(\mathbf{v}) \ &= 2H\tilde{\boldsymbol{\eta}}(\mathbf{u}) \star \mathbf{B}(\mathbf{v}),
\end{aligned}
\tag{14}
$$

cf. $\tilde{\eta}$ and $\tilde{\sigma}$ in Eq. (9). The adjoint SSA equations are

$$
\begin{aligned}
&\psi_t + \mathbf{u} \cdot \nabla\psi + 2\eta\mathbf{B}(\mathbf{u}) : \mathbf{D}(\mathbf{v}) - \rho g H \nabla \cdot \mathbf{v} + \rho g \mathbf{v} \cdot \nabla b = F_h, \ \text{in } \omega, \\
&\psi(\mathbf{x}, T) = 0, \ \text{in } \omega, \quad \psi(\mathbf{x}, t) = 0, \ \text{on } \gamma_w, \\
&\nabla \cdot \tilde{\varsigma}(\mathbf{v}) - Cf(\mathbf{u})(\mathbf{I} + \mathbf{F}_\omega(\mathbf{u}))\mathbf{v} - H\nabla\psi = -F_{\mathbf{u}}, \quad \text{in } \omega, \\
&\mathbf{t} \cdot \tilde{\varsigma}(\mathbf{v})\mathbf{n} = -C_\gamma f_\gamma(\mathbf{t} \cdot \mathbf{u})(1 + F_\gamma(\mathbf{t} \cdot \mathbf{u}))\mathbf{t} \cdot \mathbf{v}, \ \text{on } \gamma_g, \quad \mathbf{t} \cdot \tilde{\varsigma}(\mathbf{v})\mathbf{n} = 0, \ \text{on } \gamma_w, \\
&\mathbf{n} \cdot \mathbf{v} = 0, \ \text{on } \gamma.
\end{aligned}
\tag{15}
$$

Compared to Eq. (8), the advection equation depends on $\mathbf{v}$ and the influence of $\psi$ in the stress equation is different in Eq. (15). With a Weertman friction law Eq. (6), the terms $\mathbf{F}_\omega$ and $F_\gamma$ in the adjoint basal friction and the lateral friction in Eq. (15) are

$$\mathbf{F}_\omega(\mathbf{u}) = \frac{m-1}{\mathbf{u} \cdot \mathbf{u}}\mathbf{u} \otimes \mathbf{u}, \quad F_\gamma = m - 1.$$



The friction coefficients on the base and the lateral sides are perturbed by $\delta C$ and $\delta C_\gamma$ and the topography is perturbed by $\delta b$ in the SSA model. Then the perturbation $\delta \mathcal{F}$ in the observation $\mathcal{F}$ in Eq. (7) is (Cheng and Lötstedt (2019))

$$
\delta \mathcal{F} = \int_0^T \int_\omega (2\eta \mathbf{B}(\mathbf{u}) : \mathbf{D}(\mathbf{v}) + \rho g \mathbf{v} \cdot \nabla h + \nabla \psi \cdot \mathbf{u}) \, \delta b - f(\mathbf{u}) \mathbf{u} \cdot \mathbf{v} \, \delta C \, \mathrm{d}\mathbf{x} \, \mathrm{d}t
$$
$$
- \int_0^T \int_{\gamma_g} f_\gamma(\mathbf{t} \cdot \mathbf{u}) \mathbf{t} \cdot \mathbf{u} \, \mathbf{t} \cdot \mathbf{v} \, \delta C_\gamma \, \mathrm{d}s \, \mathrm{d}t. \tag{16}
$$

### 2.2.3 Forward and adjoint SSA in 2D

5   In the 2D model, $u_2 = 0$, derivatives with respect to $y$ vanish, and the lateral friction force is neglected, $C_\gamma = 0$. The ice domains are the grounded and floating parts $\Gamma_b = [0, x_{GL}]$ and $\Gamma_w = (x_{GL}, L]$ where $x_{GL}$ is the position of the grounding line. The friction coefficient $C$ is positive on $\Gamma_b$ and $C = 0$ on $\Gamma_w$. The forward and adjoint equations in 2D are derived from Eq. (13) and Eq. (15) by letting $H$ and $u_1$ be independent of $y$ and taking $u_2 = 0$. The notation is simplified if we let $u = u_1$ and $v = v_1$. The forward equations follow from Eq. (13)

$$
\begin{aligned}
&h_t + (uH)_x = a, \ 0 \le t \le T, \ 0 \le x \le L, \\
&h(x,0) = h_0(x), \ h(0,t) = h_L(t), \\
&(H\eta u_x)_x - Cf(u)u - \rho g H h_x = 0, \ 0 \le x \le L, \\
&u(0,t) = u_L(t), \ u(L,t) = u_c(t).
\end{aligned} \tag{17}
$$

Assume that $u > 0$ and $u_x > 0$. There is an inflow of ice with speed $u_L$ to the left and a calving rate $u_c$ at $x = L$. The viscosity in Eq. (11) is simplified to $\eta = 2A^{-1/n} u_x^\nu$. The friction term is $Cf(u)u = Cu^m$ with the Weertman law in Eq. (6). The adjoint variables $v$ and $\psi$ satisfy the adjoint equations in 2D

$$
\begin{aligned}
&\psi_t + u\psi_x + (\eta u_x - \rho g H)v_x + \rho g b_x v = F_h, \\
&0 \le t \le T, \ 0 \le x \le L, \\
&(\tfrac{1}{n} H\eta v_x)_x - Cmf(u)v - H\psi_x = -F_u, \\
&\psi(x,T) = 0, \ \psi(L,t) = 0, \ v(0,t) = 0, \ v(L,t) = 0,
\end{aligned} \tag{18}
$$

15   obtained from Eq. (14) and Eq. (15) or derived from Eq. (17) with equal result.

Perturbations $\delta b$ and $\delta C$ in the topography and the friction coefficient propagate to the surface as in Eq. (16)

$$
\delta \mathcal{F} = \int_0^T \int_0^L (\psi_x u + v_x \eta u_x + v \rho g h_x) \, \delta b - v f(u) u \, \delta C \, \mathrm{d}x \, \mathrm{d}t. \tag{19}
$$

### 2.2.4 Discretized relations in 2D

In order to simplify the notation, only a 2D steady state problem for the SSA model is considered here but the analysis is
20   applicable to 3D steady state problems as well as time-dependent problems with the FS or SSA models.



The time independent perturbation of $\mathcal{F}$ in Eq. (19) for the steady state solution is rewritten with $F_u = \delta(x - x_*)$ and weights $w_{ub}$ and $w_{uC}$

$$
\begin{aligned}
\delta u(x_*) &= \delta \mathcal{F} = \int_0^L w_{ub}\delta b + w_{uC}\delta C\,\mathrm{d}x, \\
w_{ub}(x_*, x) &= \psi_x u + v_x \eta u_x + v\rho g h_x,\ w_{uC}(x_*, x) = -vf(u)u.
\end{aligned}
\tag{20}
$$

The weights $w_{ub}$ and $w_{uC}$ in Eq. (20) depend on both $x_*$ and $x$. When $h$ is observed the perturbation is

$$
5\quad \delta h(x_*) = \int_0^L w_{hb}\delta b + w_{hC}\delta C\,\mathrm{d}x,
\tag{21}
$$

where the weights $w_{hb}$ and $w_{hC}$ have the same form as in Eq. (20) but with different $\psi$ and $v$.

The relation is discretized by observing $u$ at equidistant $x_{*i}$, $i = 1, 2, \ldots, M$, with $x_{*,i+1} - x_{*i} = \Delta x_*$ and perturbing $b$ and $C$ at $x_j$, $j = 1, 2, \ldots, N$, with $x_{j+1} - x_j = \Delta x$. The integral in Eq. (20) is computed by the trapezoidal rule to have

$$
\begin{aligned}
\delta u(x_{*i}) &= \sum_{j=1}^N \mu_j(w_{ub}(x_{*i}, x_j)\delta b(x_j) + w_{uC}(x_{*i}, x_j)\delta C(x_j))\,\Delta x, \\
\mu_1 &= 0.5,\ \mu_j = 1,\ j = 2, 3, \ldots, N-1,\ \mu_N = 0.5,
\end{aligned}
\tag{22}
$$

10  or in matrix form

$$
\delta \mathbf{u} = \mathbf{W}_{ub}\delta \mathbf{b} + \mathbf{W}_{uC}\delta \mathbf{C},
\tag{23}
$$

with the matrix elements

$$
\begin{aligned}
&W_{ubij} = \mu_j w_{ub}(x_{*i}, x_j),\ W_{uCij} = \mu_j w_{uC}(x_{*i}, x_j), \\
&i = 1, 2, \ldots, M,\ j = 1, 2, \ldots, N.
\end{aligned}
$$

In the same manner, there are matrices $\mathbf{W}_{hb}$ and $\mathbf{W}_{hC}$ connecting $\delta h$ with $\delta b$ and $\delta C$

$$
15\quad \delta h = \mathbf{W}_{hb}\delta \mathbf{b} + \mathbf{W}_{hC}\delta \mathbf{C}.
\tag{24}
$$

The sensitivity of $u$ to changes in $b$ and $C$ on $\omega$ is given by the singular value decomposition (SVD) of $\mathbf{W}_{ub}$ and $\mathbf{W}_{uC}$ (Golub and Loan (1989)) defined by

$$
\mathbf{W}_{ub} = \mathbf{U}_{ub}\mathbf{\Sigma}_{ub}\mathbf{V}_{ub}^T,\ \mathbf{W}_{uC} = \mathbf{U}_{uC}\mathbf{\Sigma}_{uC}\mathbf{V}_{uC}^T,
$$

where $\mathbf{U}_{ub}$ and $\mathbf{U}_{uC}$ are of size $M \times M$ and $\mathbf{V}_{ub}$ and $\mathbf{V}_{uC}$ are of size $N \times N$. They are orthogonal matrices, e.g. $\mathbf{U}_{ub}^T\mathbf{U}_{ub} = \mathbf{I}$.

20  The diagonal matrices $\mathbf{\Sigma}_{ub}$ and $\mathbf{\Sigma}_{uC}$ are of size $M \times N$ with non-negative singular values $\sigma_{ubi}$ and $\sigma_{uCi}$ in the diagonals ordered from large to small for increasing $i = 1, 2, \ldots, \min(M, N)$.

Consider a case with $\delta \mathbf{b} = \mathbf{0}$, the perturbation is simplified to $\delta \mathbf{u} = \mathbf{W}_{uC}\delta \mathbf{C}$. If $M = N$ and the smallest singular value $\sigma_{uCN} = \min_i \sigma_{uCi}$ is positive then

$$
\delta \mathbf{C} = \mathbf{W}_{uC}^{-1}\delta \mathbf{u} = \mathbf{V}_{uC}\mathbf{\Sigma}_{uC}^{-1}\mathbf{U}_{uC}^T\delta \mathbf{u}.
\tag{25}
$$





If $M > N$ with more observations of $\delta u_i$ than discrete $\delta C_j$, then $\delta \mathbf{C}$ for a given $\delta \mathbf{u}$ can be computed in the least squares sense by minimizing $\|\delta \mathbf{u} - \mathbf{W}_{uC}\delta \mathbf{C}\|$ also with the solution

$$\delta \mathbf{C} = \mathbf{V}_{uC}\mathbf{\Sigma}_{uC}^{-1}\mathbf{U}_{uC}^{T}\delta \mathbf{u}, \tag{26}$$

where $\mathbf{\Sigma}_{uC}^{-1}$ is the generalized inverse of $\mathbf{\Sigma}_{uC}$ of dimension $N \times M$ with elements $\sigma_{uCi}^{-1}$ on the diagonal and 0 elsewhere.

5 The relation between $\delta \mathbf{u}$ and $\delta \mathbf{C}$ is well behaved in Eq. (25) and Eq. (26) if all the singular values $\sigma_{uCi}$ are of similar size, but if some of them are much smaller than the other ones with $\sigma_{Ci} \ll \sigma_{C1}, i = J, J+1, \ldots, \min(M, N)$, then the relation is ill-conditioned. A large perturbation in $C$ may then result in a hardly visible perturbation at the surface and a small observed perturbation in $u$ may correspond to a large perturbation at the base. The same conclusions apply to $\mathbf{W}_{ub}$ and $\sigma_{ubi}$ in the relation between $\delta \mathbf{u}$ and $\delta \mathbf{b}$ and to the sensitivity matrices $\mathbf{W}_{hb}$ and $\mathbf{W}_{hC}$ when $F_h = \delta(x - x_*)$.

10 The transfer functions in Gudmundsson (2003) between perturbations in $b$ and $C$ at the base and the observations $u$ and $h$ at the top are determined by linearization and Fourier transformation in a slab geometry. The transfer function for different wave numbers corresponds to the singular values in our analysis.

## 3 Results

In the numerical experiments we use a 2D constant downward-sloping bed with an ice profile from the MISMIP benchmark 15 project in Pattyn et al. (2012). The bedrock elevation in meters is given as

$$b(x) = 720 - 778.5 \times \frac{x}{750 \text{ km}}. \tag{27}$$

The initial configuration of the ice is a steady state solution achieved by the FS model using Elmer/Ice (Gagliardini et al. (2013)) with $A = 1.38 \times 10^{-24} \text{ s}^{-1}\text{Pa}^{-3}$ with a grounding line position at $x_{GL} = 1.053 \times 10^{6}$ m shown in Fig. 1. The Weertman type friction law in Eq. (6) in the forward problem has the exponent $m = 1/3$ and a constant friction coefficient $C_0 = 7.624 \times$ 20 $10^6 \text{ m}^{-1/3}\text{s}^{1/3}\text{Pa}$. The remaining physical parameters are given in Table 1.

| Parameter | Quantity |
|---|---|
| $\rho_w = 1000 \text{ kg m}^{-3}$ | Water density |
| $\rho_i = 900 \text{ kg m}^{-3}$ | Ice density |
| $g = 9.8 \text{ m s}^{-2}$ | Acceleration of gravity |
| $n = 3$ | Flow-law exponent |
| $a = 0.3 \text{ m year}^{-1}$ | Accumulation rate |

**Table 1.** The physical parameters of the ice.

Without losing the generality in the friction law and to investigate the relation between the basal velocity and the stress, the friction law exponent in the adjoint problem is assumed to be $m = 1$ and the coefficient is calculated from the forward

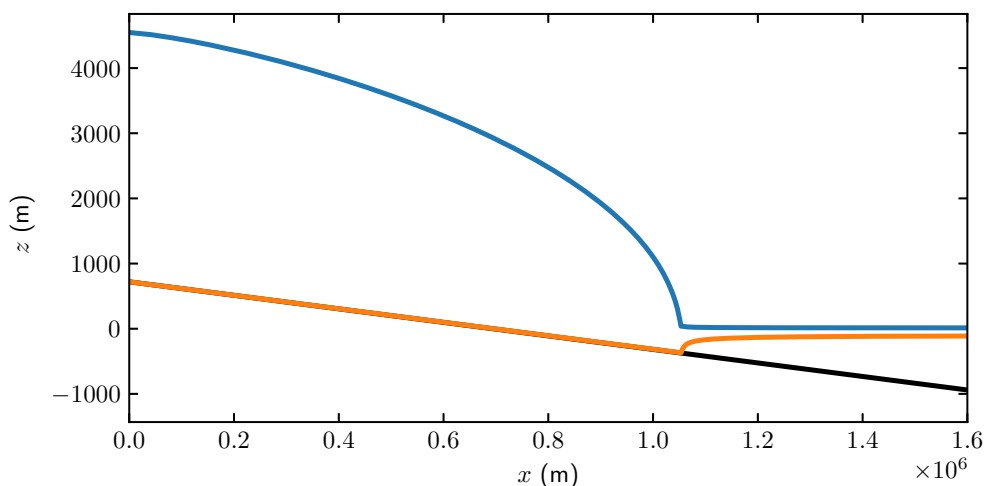

**Figure 1.** The initial ice geometry with height $h$ (blue), ice base $b$ (orange), and ocean bathymetry (black). The domains in Eq.(2) are the ice domain $\Omega$ between the blue and orange curves, the upper surface $\Gamma_s$ in blue, the lower boundary on the bedrock $\Gamma_b$ and on water $\Gamma_w$ in orange, $\Gamma_u$ at $x = 0$ and $\Gamma_d$ at $x = L = 1.6 \times 10^6$ m.

steady state solution by $C(\mathbf{x}) = C_0 \|\mathbf{u}\|^{-2/3}$. The resulting friction law becomes $Cf(\mathbf{u}) = C(\mathbf{x})$ which can be viewed as a linearization of the friction law at the steady state.

### 3.1 Full Stokes model

A vertically extruded mesh is constructed for the given geometry with mesh size $\Delta x = 1$ km yielding equidistant nodes in the

horizontal direction. The number of vertical layers is set to 20 in the whole domain. Only the grounded ice is considered in the adjoint problem and Dirichlet boundary conditions on $\mathbf{u}$ are used for the lateral boundaries $\Gamma_d$ and $\Gamma_u$ at the grounding line $x = x_{GL}$ and the ice divide $x = 0$.

The forward and adjoint FS problems are solved using the finite element code Elmer/Ice (Gagliardini et al. (2013)) with P1-P1 quadrilateral element and Galerkin Least Squares stabilization for the Stokes equation and a bubble stabilization (Baiocchi

et al. (1993)) for the adjoint advection equation. The feature to solve the adjoint time dependent equations has been added to Elmer/Ice. The Dirac delta is approximated by a linear basis function with the amplitude $1/\Delta x$.

The time stepping scheme for the forward and adjoint transient problems is the implicit Euler method with a constant time step $\Delta t = 1$ year. The adjoint equation is solved backward in time from the final time $t = T$ to $t = 0$. The steady state of the adjoint equations is computed by neglecting the time derivative term in the adjoint surface equation Eq. (8) and solving the

corresponding linear system of equations for $\psi$ and $\mathbf{v}$.



Both transient and steady state simulations are run with pointwise observations of the horizontal velocity $u_1$ and surface elevation $h$ at different $x_*$ positions on the top surface. The time interval for the transient solutions is $[0, 1]$ covered by one forward timestep $\Delta t$ from 0 to 1 and one backward timestep from 1 to 0.

The multiplier $\psi$ only acts as the amplitude of the external force on $\Gamma_s$ and $\mathbf{h}$ is an approximate normal vector pointing inward

5 on $\Gamma_s$ in the adjoint FS equation Eq. (8). The size of $\psi\mathbf{h}$ is several orders of magnitude smaller than 1, the coefficient in front of $\delta(x - x_*)$ in $F_{\mathbf{u}}$. Consequently, in the $u_1$-response case, the adjoint solution $\mathbf{v}$ is mainly influenced by the observation function $F_{\mathbf{u}}$. However, in the $h$-response case with $F_{\mathbf{u}} = 0$, the adjoint solution $\mathbf{v}$ is determined by $\psi\mathbf{h}$ and the solution would be $\mathbf{v} = \mathbf{0}$ if we did not solve the adjoint advection equation for $\psi$.

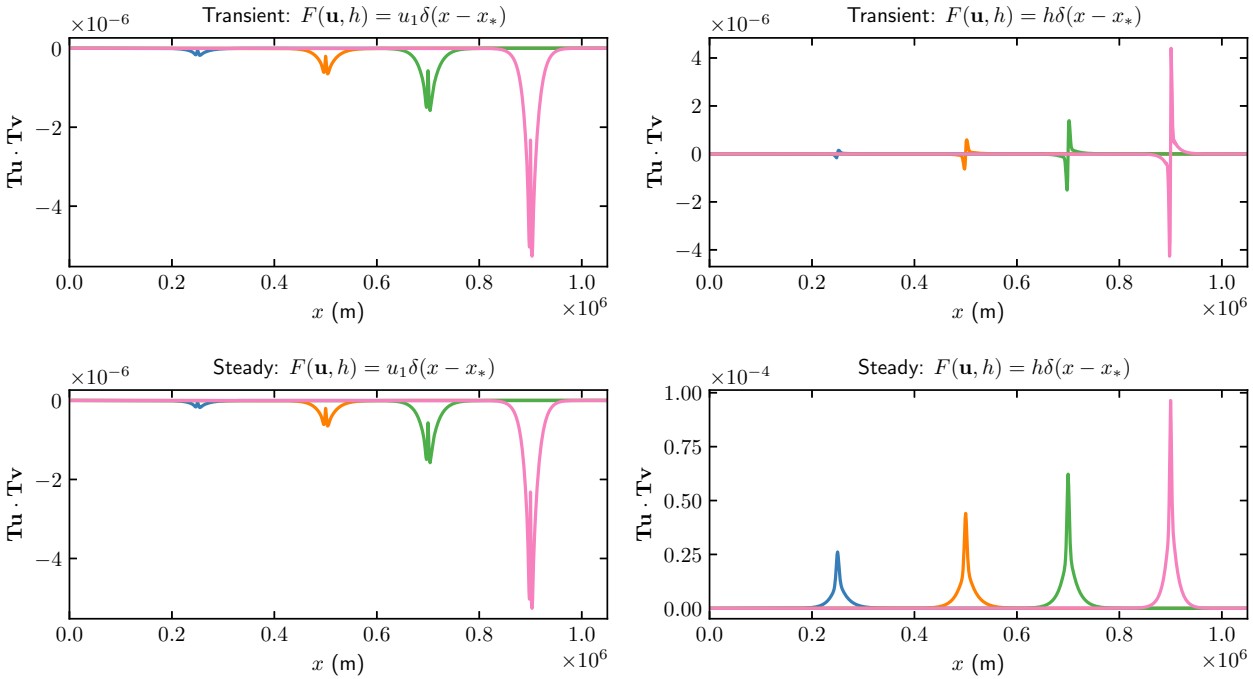

**Figure 2.** Comparison of the weights $\mathbf{Tu} \cdot \mathbf{Tv}$ in Eq. (10) for perturbations $\delta C$ at different observation points $x_* = 0.25 \times 10^6, 0.5 \times 10^6, 0.7 \times 10^6$ and $0.9 \times 10^6$ (blue, orange, green, and pink). *Upper panels*: transient simulations; *lower panels*: steady states. *Left panels*: $w_{uC}$ with pointwise $\mathbf{u}$ response; *right panels*: $w_{hC}$ with pointwise $h$ response.

The adjoint solutions $v_1$ at $\Gamma_b$ of all the four cases are concentrated at the observation points. The vertical component $v_2$ shares

10 the same feature as $v_1$ due to the boundary condition $\mathbf{n} \cdot \mathbf{v} = 0$ on $\Gamma_b$. Therefore, the weights $\mathbf{Tu} \cdot \mathbf{Tv}$ in Fig. 2 are also confined to the neighborhood of $x_*$. The negative weights obtained in the $u_1$-response cases imply that an increase in the basal friction coefficient results in a decrease of the surface velocity. The amplitude of the weights grows rapidly toward the grounding line in all four cases in the figure. In fact, the contribution of the weight function to the observed variables $u_1$ can be viewed as a



convolution of the perturbation in $C(x)$ with a narrow Gaussian $w_{uC}(x_*, x)$ in Eq. (20) after a proper scaling in the left panels of Fig. 2.

The amplitude of the perturbation at the surface depends on the wavelength $\lambda$ of the perturbation at the base. The shorter $\lambda$ is, the smaller the amplitude is. Introduce a stationary perturbation $\delta C(x) = \epsilon C_0 \cos(2\pi(x - x_*)/\lambda)$ with a constant $C_0$ and a small $\epsilon \ll 1$. Then the change in the steady state solution $u_1$ at the surface is according to Eq. (10)

$$\delta u_1(x_*, \lambda) = \int_0^L \epsilon C_0 \mathbf{Tu} \cdot \mathbf{Tv} \cos\left(\frac{2\pi(x - x_*)}{\lambda}\right) \, \mathrm{d}x. \tag{28}$$

The same relation holds for $\delta h(x_*)$ but with a different $\mathbf{v}$. Let $\varrho$ be a measure of the width of the weight function for the steady state in Fig. 2 which is about $10^5$. When $\lambda$ is large compared to $\varrho$ then

$$\delta u_1(x_*, \lambda) \approx \delta u_{1,\infty}(x_*) = \lim_{\lambda \to \infty} \delta u_1(x_*, \lambda) = \epsilon C_0 \int_0^L \mathbf{Tu} \cdot \mathbf{Tv} \, \mathrm{d}x, \tag{29}$$

which is a constant value for long $\lambda$, and the perturbation can be observed at the surface. If the wavelength of the basal perturbation is short compared to $\varrho$, then it is damped before it reaches the surface and the effect of $\delta C$ on $u_1$ and $h$ is small. In Fig. 3, $\delta u_1(x_*, \lambda)$ and $\delta u_{1,\infty}(x_*)$ are compared at $x_* = 0.9 \times 10^6$. When $\lambda > \varrho$ then $\delta u_1(x_*, \lambda) \approx \delta u_{1,\infty}(x_*)$. Suppose that $\lambda = 2 \times 10^4$. Then $\delta u_1(x_*, \lambda)$ is about $0.02\delta u_{1,\infty}(x_*)$ and probably hard to observe and $\delta h(x_*, \lambda) \approx 0.2\delta h_\infty(x_*)$. Similar conclusions are drawn theoretically in Gudmundsson (2003) using Fourier analysis and experimentally in Sun et al. (2014).

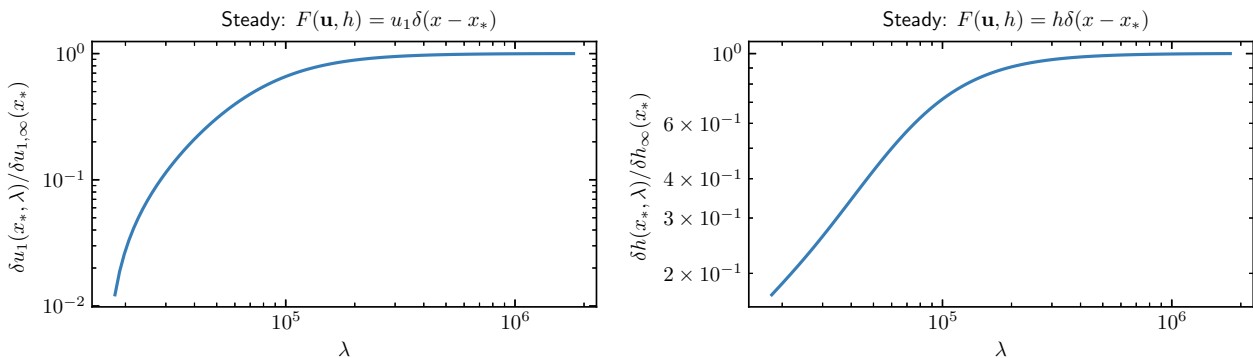

**Figure 3.** The response at $\Gamma_s$ with different wavelengths $\lambda$ in the perturbation of $C$ in Eq. (28). *Left panel*: $\delta u_1(x_*, \lambda)/\delta u_{1,\infty}(x_*)$; *right panel*: $\delta h(x_*, \lambda)/\delta h_\infty(x_*)$.

We perform a pair of experiments to compare the results from perturbing the forward equation and the prediction by the adjoint solutions. A relative 1% perturbation $\delta C(x)$ is added at $x \in [0.9, 1.0] \times 10^6$ m to the friction coefficient $C(x)$. The differences between the forward FS solutions with and without the perturbation after one year are shown in Fig. 4 marked as 'perturbed'. The 'predicted' perturbations are computed from the solutions of the adjoint equation by varying $x_*$ along the $x$-axis and



inserting into Eq. (10). Each red dot in Fig. 4 corresponds to one single observation at $x_*$. Both the $u_1$ and $h$ predictions are in good agreement with the forward perturbations.

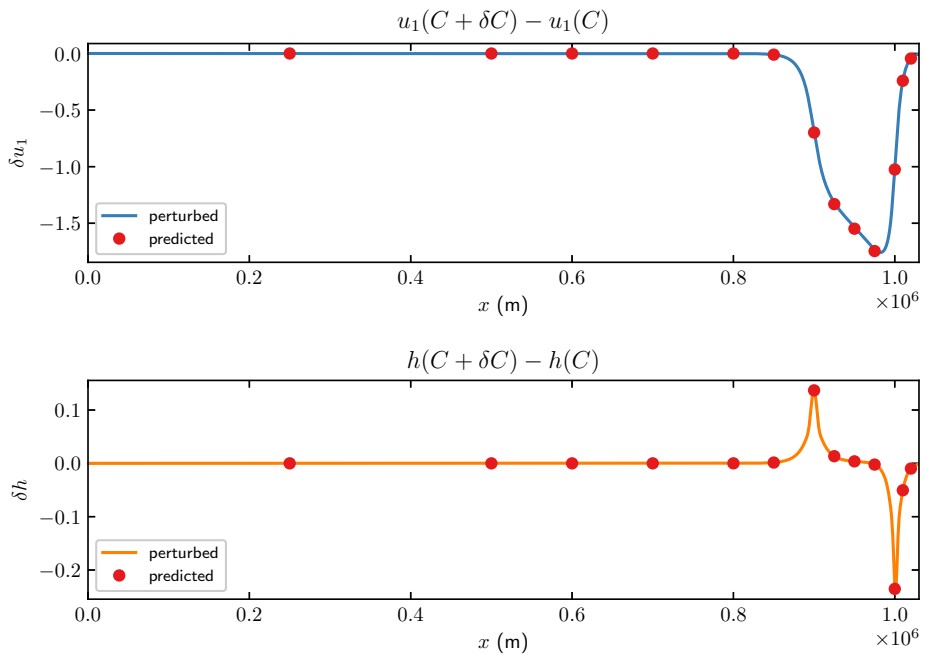

**Figure 4.** The changes on the horizontal velocity $u_1$ (upper panel) and surface elevation $h$ (lower panel) after one year with $1\%$ perturbation on $C(x)$ at $x \in [0.9, 1.0] \times 10^6$ m. Solid lines are the differences between the steady state and perturbed transient solutions in Eq. (5). Red dots are the estimated perturbation using Eq. (10).

## 3.2  SSA

The same MISMIP benchmark experiment as in Sect. 3.1 is solved by the SSA on a one dimensional uniform grid with mesh
5  size $\Delta x = 1$ km using standard finite difference methods implemented in MATLAB. The time derivatives are discretized by the implicit Euler method with a constant time step $\Delta t = 1$ year as in Sect. 3.1. An upwind scheme is used for the spatial derivatives in the forward and adjoint advection equations to stabilize the numerical solutions. Replacing the Dirac delta with a Gaussian of a few grid points wide in order to smoothen the observation function and avoid numerical oscillations in the solution has no major effect on the solutions.
10  The numerical solution of the forward SSA equations Eq. (17) is compared to the analytical approximations in the Appendix Eq. (A1) in Fig. 5. The detailed derivation of the analytical solutions in the Appendix are found in Cheng and Lötstedt (2019). The analytical approximation of $u$ is poor to the right of $x_{GL}$ for the floating ice in Fig. 5 but we are only interested in the solution on the ground. The reason for the error in the analytical solution of $u$ is that $H$ is assumed to be constant for $x > x_{GL}$.



The analytical solution for $H$ catches the fast decrease when $x$ approaches $x_{GL}$ from the left. Another solution for $x > x_{GL}$ is found in Greve and Blatter (2009) assuming that the thickness depends linearly on $x$.

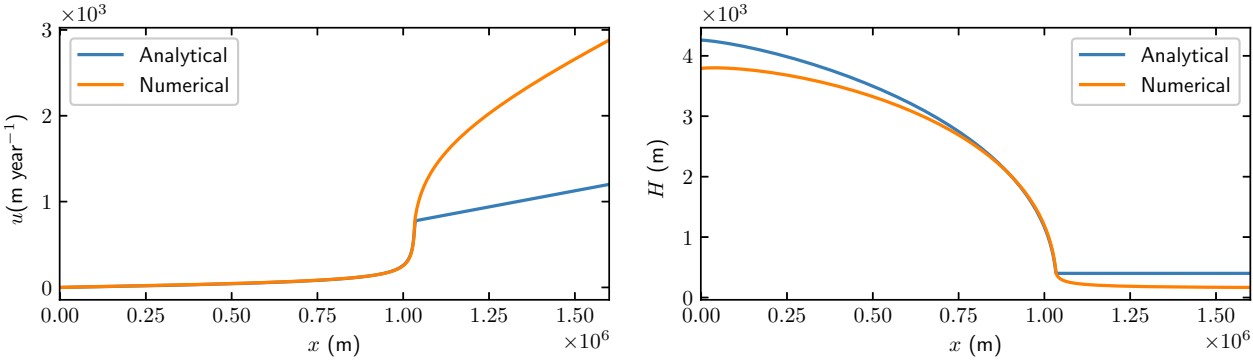

**Figure 5.** Comparison of the steady state numerical solutions of the SSA velocity $u$ and the thickness $H$ in Eq. (17) (orange) and the analytical solutions in Eq. (A1) (blue).

The weight functions $w_{uC}$ and $w_{hC}$ in Fig. 6 have the same non-zero pattern as $v$ since they are equal to $-vu^m$ in Eq. (20). Each one of these weights $w_{uC}$ or $w_{hC}$ corresponds to the sensitivity of the observation at $x_*$ with respect to the change in

$C(x)$ which is one row in the weight matrices $\mathbf{W}_{uC}$ or $\mathbf{W}_{hC}$ in Eq. (23) and Eq. (24). The analytical weight functions in Eq. (A3) and Eq. (A5) at $x_* = 0.7 \times 10^6$ m are included in the steady state for comparison. In the transient SSA simulations, the sensitivity is similar to those in the adjoint FS solutions in Fig. 2 increasing towards the grounding line. This increased sensitivity is also noted in Kyrke-Smith et al. (2018); Leguy et al. (2014). However, in the steady state cases, the weight functions indicate only an upstream effect of $C(x)$. In other words, the perturbation in $C(x)$ at point $x$ can only influence the

steady state solutions to the left of this point. This is true as long as the effect of the grounding line migration is neglected. The $\delta C$ weights for $u$ responses are all negative implying that an increase of $C$ leads to decrease of $u$, but the steady state surface elevation $h$ rises when $C$ is increased. The weights for the transient problem have similar shape for the FS and SSA models in Figs. 2 and 6.

The weight functions $w_{ub}$ and $w_{hb}$ for $\delta b$ are localized at the observation position $x_*$ in all the four cases in Fig. 7 which implies

that the inverse problems may be well posed. The black dashed lines in the two lower panels are the analytical expressions of the weight functions at $x_* = 0.7 \times 10^6$ m in Eq. (A3) and Eq. (A5) with a hat function of width $2\Delta x$ at the base to approximate the Dirac delta. The analytical solutions almost coincide with the numerical solutions. The steady state weight functions are non-zero to the right of $x_*$. There is a detailed view of the steady state $\delta b$ weights for $x > x_*$ in Fig. 8. The weights of $\delta b$ have similar structures as the $\delta C$ weights. The analytical solutions in Eq. (A3) and Eq. (A5) suggest that $w_{ub}/w_{uC} \approx w_{hb}/w_{hC} \approx$

$(m+1)C/H$ for $x \neq x_*$.

The inverse problem of the steady state for the friction coefficient may not be well posed since the weights are all positive from $x_*$ to $x_{\mathrm{GL}}$. This is verified by checking the singular values of the sensitivity matrices $\mathbf{W}_{uC}$ and $\mathbf{W}_{hC}$ in Fig. 9 where the



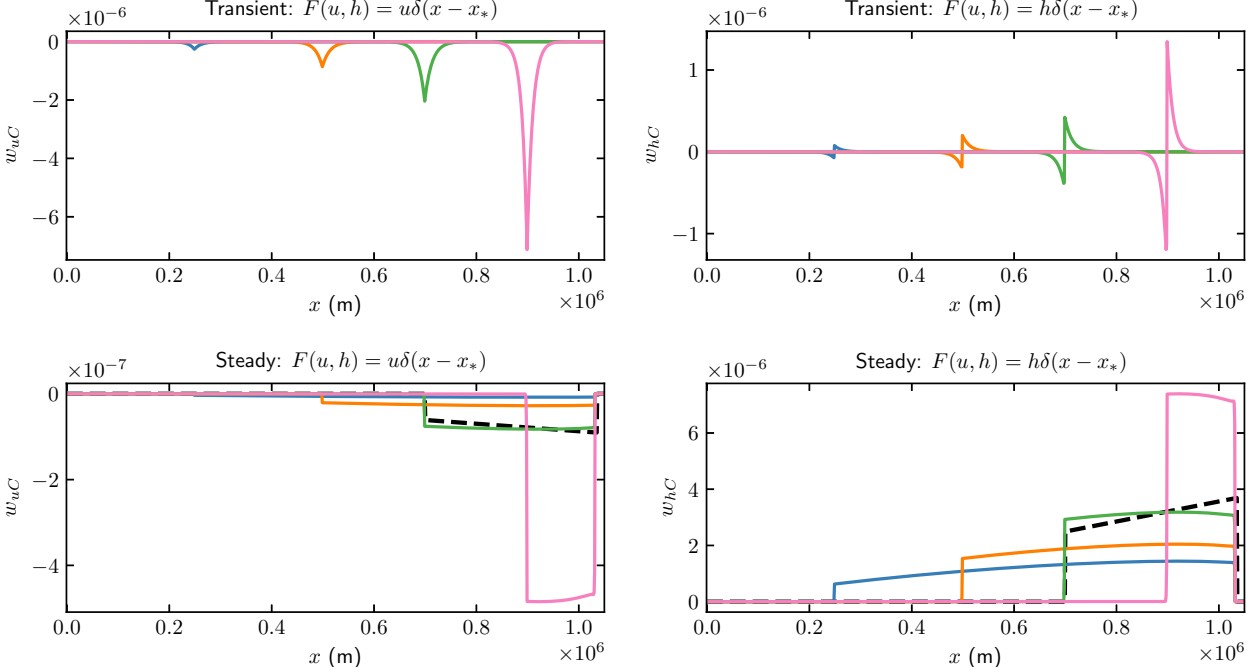

**Figure 6.** Comparison of the weights $w_{uC}$ and $w_{hC}$ in Eq. (19) for perturbations $\delta C$ with $m = 1$ at different observation points $x_* = 0.25 \times 10^6, 0.5 \times 10^6, 0.7 \times 10^6$ and $0.9 \times 10^6$ (blue, orange, green, and pink). The black dashed line in the lower panels are $w_{uC}$ and $w_{hC}$ computed from the analytical solutions of $u$ in Eq. (A1) and $v$ in Eq. (A2) and Eq. (A4) at $x_* = 0.7 \times 10^6$. *Upper panels*: transient simulations; *lower panels*: steady states. *Left panels*: $w_{uC}$ for pointwise **u** response; *right panels*: $w_{hC}$ for pointwise $h$ response.

largest and smallest singular values of $\mathbf{\Sigma}_{uC}$ are $10^{-4}$ and $10^{-12}$ with a large quotient $\sigma_{uC1}/\sigma_{uCN}$ and the span of the singular values of $\mathbf{\Sigma}_{hC}$ is from $10^{-4}$ to $10^{-8}$ (which is better).

The singular values of the sensitivity matrices $\mathbf{W}_{ub}$ and $\mathbf{W}_{hb}$ in Fig. 9 are in the interval $10^{-4}$ to $10^{-7}$ from large to small. They are better conditioned than the sensitivity matrices for $C$. In particular, $\mathbf{\Sigma}_{hb}$ (in pink-red) in the $h$-response case has the

5 lowest variation of the singular values. The inverse problem of solving for the topography $b$ from the surface elevation $h$ in the steady state setup is a well-posed problem compared to inferring $C$ from $u$.

The same perturbation on $C(x)$ as in Fig. 4 is imposed in the SSA simulations. The perturbed solutions after one year and 15,000 years (which is close to a steady state) are computed with the forward equations and then the reference solutions at the steady state without any perturbation are subtracted. This difference is compared to the perturbations obtained with the adjoint

10 equations as in Fig. 4. In the one year perturbation experiment in Fig. 10, the transient weight functions in the upper panels in Fig. 6 are used for the sensitivity estimates. The weight functions in the upper panels of Fig. 7 predict the response in Fig. 11.


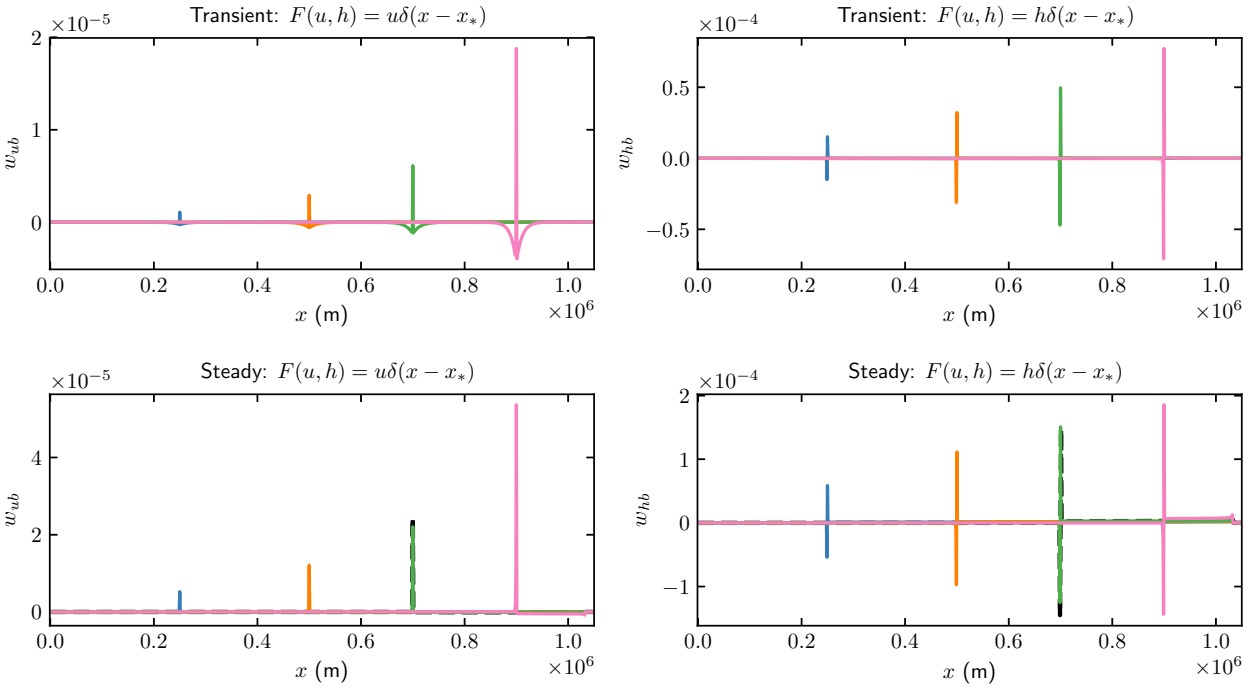

**Figure 7.** Comparison of the weights $w_{ub}$ and $w_{hb}$ in Eq. (19) for perturbations $\delta b$ at different observation points $x_* = 0.25 \times 10^6, 0.5 \times 10^6, 0.7 \times 10^6$ and $0.9 \times 10^6$ (blue, orange, green, and pink). The black dashed line in the lower panels are the weights of $\delta b$ in Eq. (A3) and Eq. (A5) at $x_* = 0.7 \times 10^6$. *Upper panels*: transient simulations; *lower panels*: steady states. *Left panels*: $w_{ub}$ for pointwise **u** response; *right panels*: $w_{hb}$ for pointwise $h$ response.

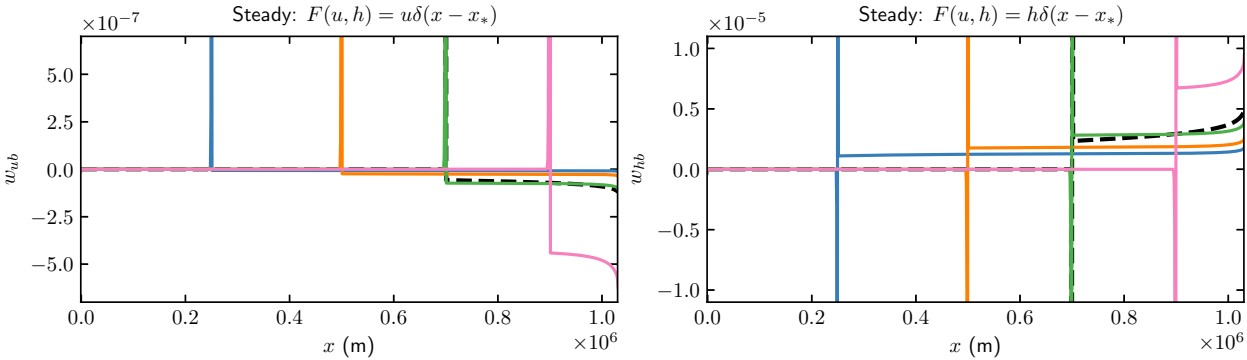

**Figure 8.** A close-up view of the steady state weights in the lower panels of Fig. 7.

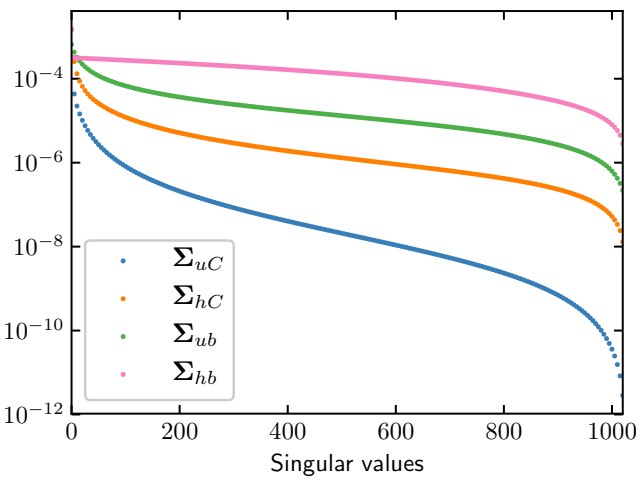

**Figure 9.** The singular values of the transfer matrices $\mathbf{W}_{uC}$, $\mathbf{W}_{hC}$, $\mathbf{W}_{ub}$ and $\mathbf{W}_{hb}$.

The corresponding comparisons for the steady state problem are made in Figs. 12 and 13 with the weights in the lower panels of Figures 6 and 7. The analytical solutions of the steady state perturbations from (A3) and (A5) are shown with black dashed lines in these two figures.

The rapid change of $\delta h$ in Figs. 10 and 11 is explained by the shape of the weight functions in the upper right panels of Figs. 6 and 7. The weights can be approximated by $-\theta(x,t)\delta'(x-x_*)$ for some $\theta > 0$. Then the surface response will be

$$\delta h(x_*) = \int_0^T \int_0^L -\theta(x,t)\delta'(x-x_*)\delta C(x)\,\mathrm{d}x\mathrm{d}t = \int_0^T (\theta\delta C)'(x_*,t)\,\mathrm{d}t,$$

where $\delta C$ jumps discontinuously at $x = 0.9 \times 10^6$ and $x = 1.0 \times 10^6$. The same phenomenon is found for FS in Fig. 4 with an explanation in Fig. 2.

The perturbations $\delta u$ and $\delta h$ in the steady state in Fig. 12 have discontinuous derivatives $\delta u_x$ and $\delta h_x$ where $\delta C$ has jumps. This is explained by the integral terms in (A3) and (A5). The discontinuities in the upper panel of Fig. 13 are caused by the jumps in $\delta b$ at $0.9 \times 10^6$ and $1.0 \times 10^6$ and the first term in (A3). The jumps in $\delta h$ in the lower panel of Fig. 13 are due to the first term in (A5).

All the predicted solutions from the adjoint SSA are in good agreement with the forward perturbation.

The solution of the adjoint equations is simplified in the comparison in Fig. 14. In MacAyeal (1993), two simplifications are made. Firstly, the adjoint viscosity $\tilde{\boldsymbol{\eta}}$ in Eq. (14) is approximated by the forward viscosity $\eta$ in Eq. (11). The factor $1/n$ in the viscosity in the 2D stress equation Eq. (18) is then replaced by $1$. Secondly, the thickness $H$ is fixed and the advection equation for $\psi$ is not solved, which is equivalent to $\nabla\psi = \mathbf{0}$ in the adjoint stress equation in Eq. (15). Perturbations are introduced in $C$ and $u$ is observed for the transient case as in Fig. 10. The perturbed forward solutions are compared to the predicted



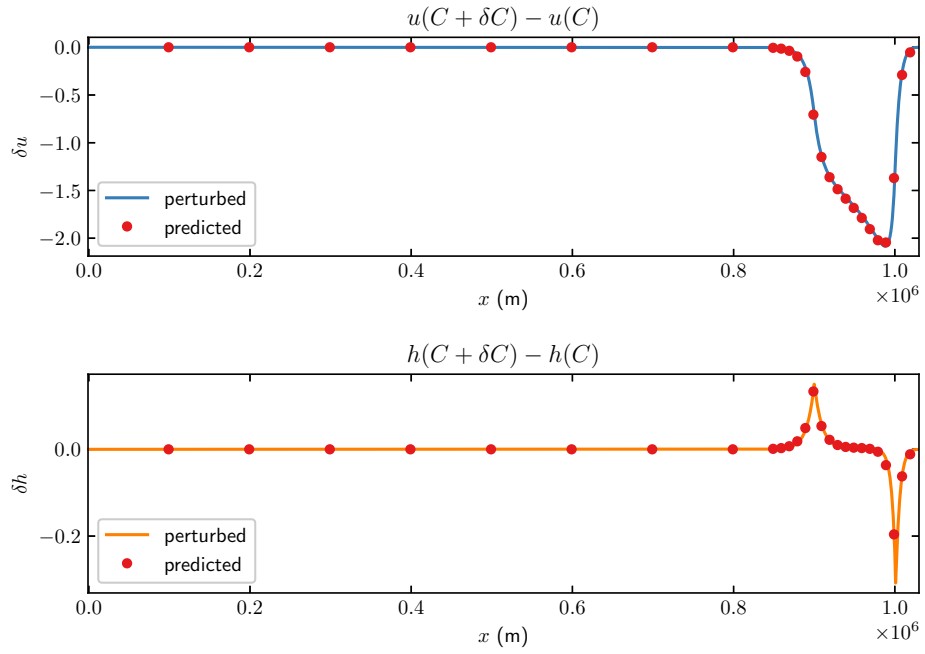

**Figure 10.** The changes in the horizontal velocity $u$ (upper panel) and surface elevation $h$ (lower panel) after one year with 1% perturbation of $C(x)$ in $x \in [0.9, 1.0] \times 10^6$ m. Solid lines are the differences between the steady state and the perturbed solutions in Eq. (13). Red dots represent the estimated perturbation using Eq. (15).

perturbations by the simplified adjoint SSA systems in Fig. 14, where the forward viscosity $\eta$ is used in both cases. In the upper panel of Fig. 14, the two equations of $\psi$ and $v$ are solved. In the lower panel, the advection equation of $\psi$ is excluded from the system. The differences are small in this case compared to the full adjoint solution used in Fig. 10. The reason is that $\psi, \psi_x$, and $H\eta u_x$ are small in Eq. (18).

5   The singular values of the transfer matrices corresponding to the two simplifications are shown in Fig. 15 where the two transfer matrices are denoted by $\widetilde{\mathbf{W}}_{uC}$ for the system coupling $\psi$ and $v$ and by $\widehat{\mathbf{W}}_{uC}$ for the adjoint equation without $\psi$ with a fixed $H$. The singular values in $\widetilde{\boldsymbol{\Sigma}}_{uC}$ are similar to those in $\boldsymbol{\Sigma}_{uC}$ in Fig. 9 since the influence of the adjoint viscosity on the system is almost negligible. The transfer matrix $\widehat{\mathbf{W}}_{uC}$ has a better conditioning than $\widetilde{\mathbf{W}}_{uC}$, although it is still worse than the best cases in Fig. 9. This implies that the inversion of steady state SSA without the height coupling may be an ill-posed problem.

10   Regularization is necessary penalising oscillatory behavior at the base as in Gagliardini et al. (2013); Petra et al. (2012).

## 4   Discussion

A few issues are discussed here related to the control method for estimating the parameter sensitivity.


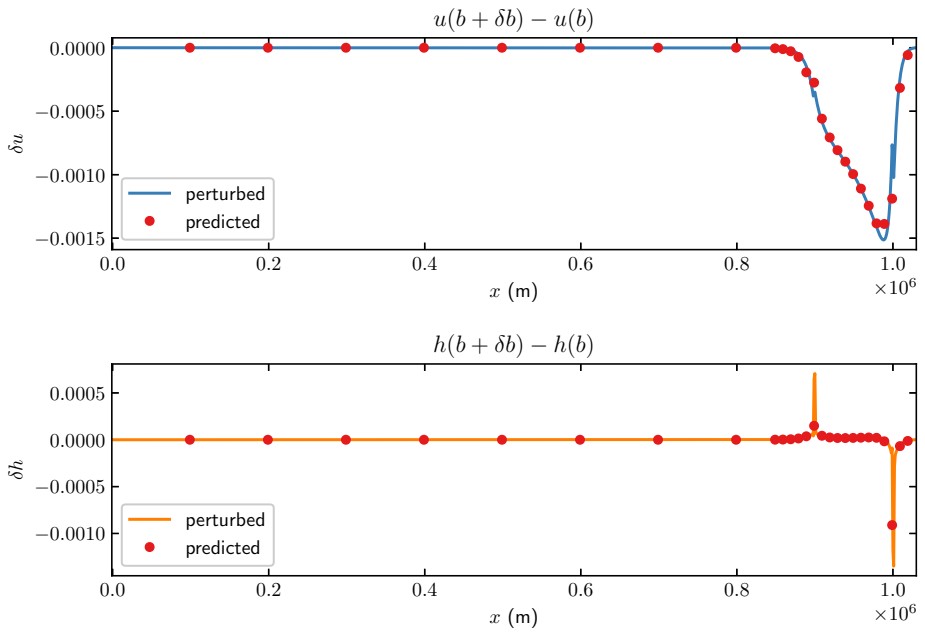

**Figure 11.** The changes in the horizontal velocity $u$ (upper panel) and surface elevation $h$ (lower panel) after one year with 0.01 m perturbation of $b(x)$ in $x \in [0.9, 1.0] \times 10^6$ m. Solid lines are the differences between the steady state and the perturbed solutions in Eq. (13). Red dots represent the estimated perturbation using Eq. (15).

We solve the FS adjoint problem only one step backward in time to verify the numerical method due to limitations of the current framework of Elmer/Ice. It is possible but more complicated and expensive to solve the adjoint problem numerically for a large number of time steps $K$. This requires storing all the forward solutions $(\mathbf{u}^i, p^i, h^i)$, $i = 1, 2, \ldots, K$, to be able to compute the adjoint solutions $(\mathbf{v}^i, q^i, \psi^i)$, $i = K, K-1, \ldots, 1$, which may be prohibitive in 3D. Since the data to be stored

5   in the SSA model is one dimension lower, we are able to solve the adjoint problem backward in time for any number of $K$. However, for a fair comparison, we show the results for one time step with SSA in this paper.

The solutions of the horizontal velocity $u$ and the height $h$ with perturbations in $C$ in the transient FS and SSA models are similar in Figures 4 and 10. The weights in the upper panels in Figures 2 and 6 are similar, too. The solutions to the forward equations are also close in the chosen MISMIP configuration. The reason is that the sliding on the ground in the FS model is

10   considerable, making SSA a good approximation of FS.

There are many discussions regarding the choice of friction laws, see e.g. Gladstone et al. (2017); Tsai et al. (2015); Brondex et al. (2017). However, assuming a spatial variability of the friction coefficient $C(\mathbf{x})$ with a linear relation between the basal stress and velocity makes this numerical study independent of the friction law. The friction coefficient can be viewed as a linearization of the friction law and a post-processing procedure can retrieve the corresponding friction law.





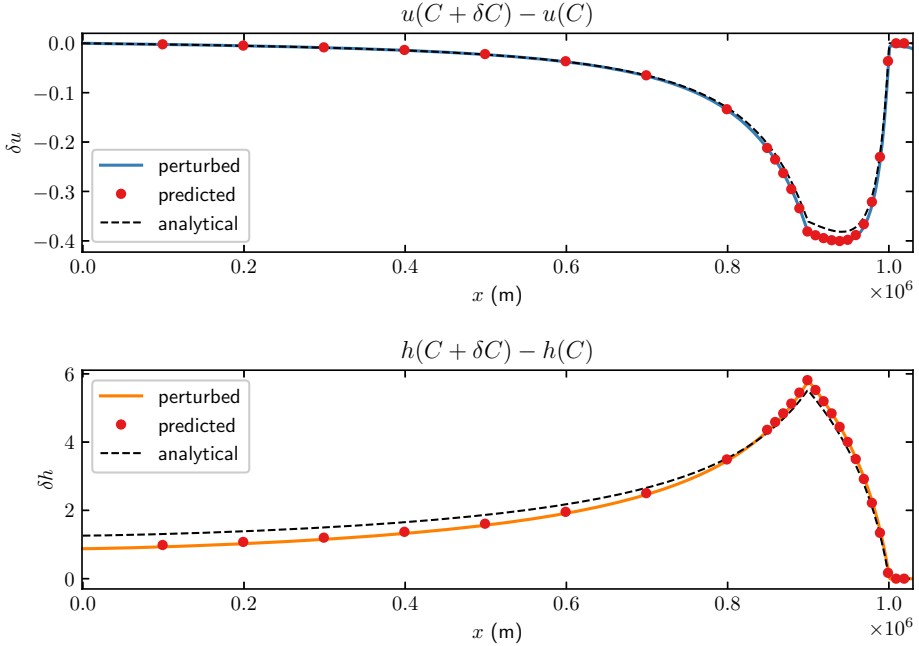

**Figure 12.** The changes in the horizontal velocity $u$ (upper panel) and surface elevation $h$ (lower panel) after 15000 years (close to the steady state) with $1\%$ perturbation of $C(x)$ in $x \in [0.9, 1.0] \times 10^6$. Solid lines are the differences between the steady state and perturbed solutions in Eq. (13). Red dots represent the estimated perturbation using Eq. (15).

The transfer relation $\mathbf{W}_{uC}$ between small perturbations of the friction coefficient $C$ at the ice base and the perturbation of the horizontal velocity $u$ at the ice surface is given by Eq. (23) with $\delta b = 0$. The singular values of $\mathbf{W}_{uC}$ in Fig. 9 tell how sensitive $u$ is to changes in $C$. The transfer relation also describes how the uncertainty in $C$ is propagated to uncertainty in the velocity at the surface and how uncertainty $\delta u$ in measurements of $u$ appear as uncertainty $\delta C$ in $C$ Eq. (26), see Smith (2014).

5    The transfer relation is computed by solving the forward problem once and then the adjoint problem for each one of the $M$ observations. An alternative would be to solve the forward equations first for the unperturbed solution and then perturb $C$ by $\delta C_j$ and solve the forward equations again $N$ times and subtract to find the relation between $\delta \mathbf{u}$ and $\delta C_j$. It is usually more expensive to solve the nonlinear forward equations than the linear adjoint equations. Suppose that the computational work to solve the forward problem is $\mathcal{W}_F$ and the adjoint problem is $\mathcal{W}_A$. If the forward and adjoint equations are in similar

10    form, such as the FS or SSA problem, and solving the nonlinear forward problem requires $k$ iterations where every nonlinear iteration has the same computational cost as solving the linear adjoint problem, then $\mathcal{W}_A/\mathcal{W}_F \approx 1/k$. The quotient between the work to determine the transfer relation involving the adjoint equations and the work only based on the forward equation is $(1 + M\mathcal{W}_A/\mathcal{W}_F)/(1 + N)$. Since $k \geq 1$, it is advantageous to choose the approach involving the adjoint if $M < kN$. Otherwise,





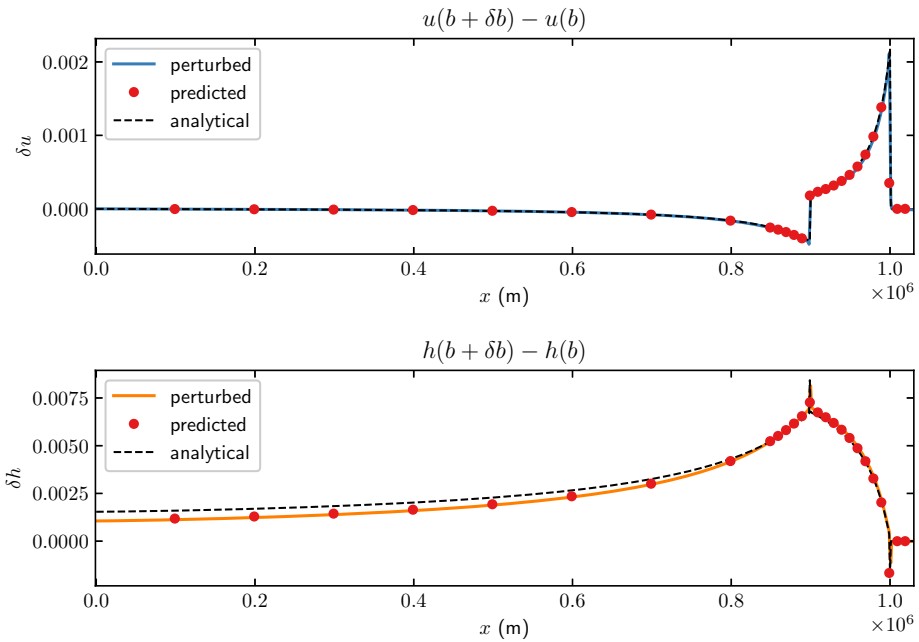

**Figure 13.** The changes in the horizontal velocity $u$ (upper panel) and surface elevation $h$ (lower panel) after 15000 years (close to the steady state) with 0.01 m perturbation of $b(x)$ in $x \in [0.9, 1.0] \times 10^6$. Solid lines are the differences between the steady state and perturbed solutions in Eq. (13). Red dots represent the estimated perturbation using Eq. (15).

solve $N + 1$ forward problems to compute $\mathbf{W}_{uC}$. In the inverse problem to find $C$ given observations of $u, h$, the functions $F_{\mathbf{u}}$ and $F_h$ are smooth and $M = 1$ in the iterative procedure to compute $C$. Solving the adjoint equations is then always favorable.

## 5 Conclusions

The perturbations $\delta u$ and $\delta h$ in the velocity $u$ and the height $h$ at the ice surface are caused by perturbations $\delta b$ and $\delta C$ in the
5  topography of the ice base $b$ and the basal friction coefficient $C$. The sensitivities $\delta u$ and $\delta h$ to $\delta b$ and $dC$ are evaluated in 2D by first solving the adjoint equations of the FS and SSA models including the advection equation for the height derived in Cheng and Lötstedt (2019). Then weight or transfer functions are determined for the relation between $\delta u$ and $\delta h$ at the surface and $\delta b$ and $\delta C$ at the base. The predictions of $\delta u$ and $\delta h$ with the weights are compared to explicit calculations of perturbed $u$ and $h$ at the surface with good agreement. It is shown in Cheng and Lötstedt (2019) that if the base perturbations are time
10  dependent then it is necessary to have time dependent weight functions to obtain the correct behavior at the top of the ice. Both the height and the stress equations and their adjoints are solved to find the weight functions here. The inverse problem at steady state to infer $C$ from observations of $u$ is usually solved for a fixed ice geometry and with only the stress equation and its adjoint, see e.g. MacAyeal (1993); Petra et al. (2012). This is possible since the adjoint height $\psi$ is small when the horizontal



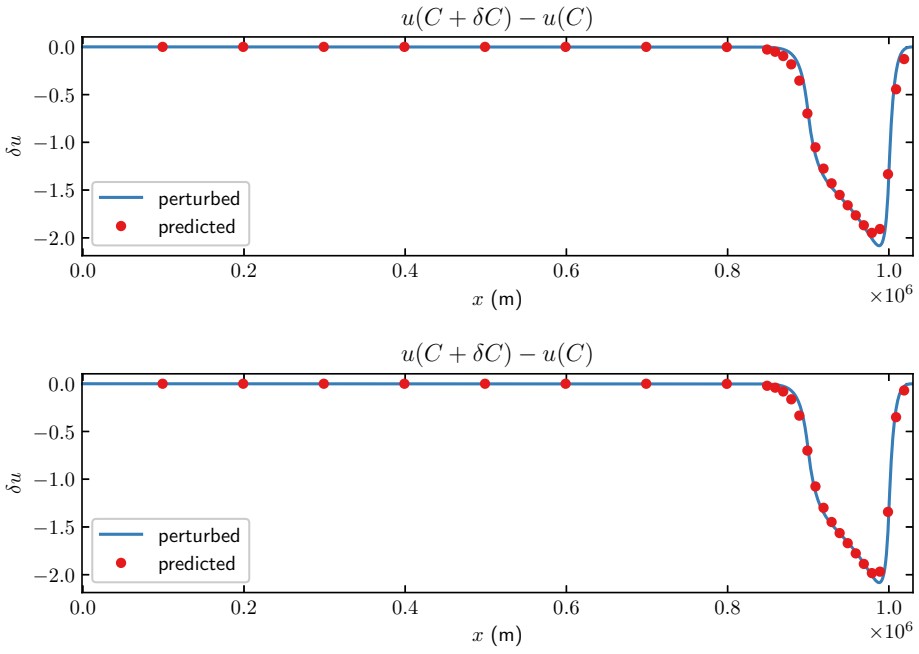

**Figure 14.** The changes in the horizontal velocity $u$ after one year with $1\%$ perturbation of $C(x)$ in $x \in [0.9, 1.0] \times 10^6$ m. Solid lines are the differences between the steady state and the perturbed solutions in Eq. (13). Red dots represent the estimated perturbation using Eq. (15). Upper panel: forward viscosity. Lower panel: without advection equation.

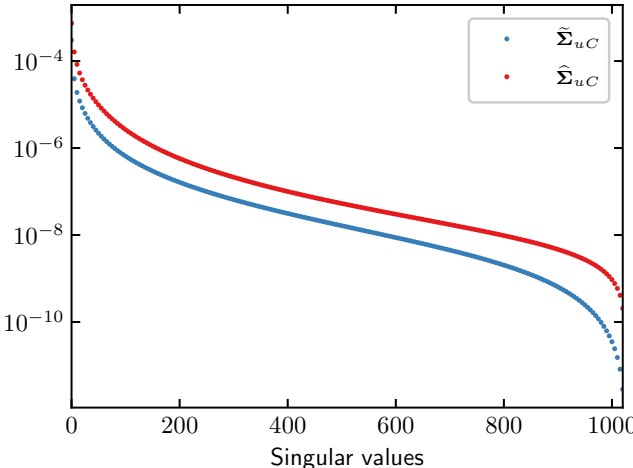

**Figure 15.** The singular values of the transfer matrices with simplifications from MacAyeal (1993). $\widetilde{\boldsymbol{\Sigma}}_{uC}$ corresponds to the forward viscosity case and $\widehat{\boldsymbol{\Sigma}}_{uC}$ is from the adjoint SSA without coupling to the $\psi$ equation.





part of $\mathbf{u}$ is observed and has little influence on $\delta \mathbf{u}$. On the contrary, if $h$ is observed then there is an important effect of $\psi$ on $\delta h$ in FS and SSA. The magnitudes of $\psi$ are different depending on whether $u$ or $h$ is observed. Simplifications of the SSA adjoint in the steady state by using the forward viscosity or ignoring the adjoint height equation have minor consequences for the predictions of $u$ with a perturbed $C$ in Fig. 14.

The sensitivity to perturbations $\delta b$ and $\delta C$ is quantified for steady state and time dependent problems with the FS and SSA models. It increases as the observation point $\mathbf{x}_*$ approaches the grounding line. This is explained by analytical expressions for SSA where the sensitivity is inversely proportional to the ice thickness $H(x_*)$. The closer we are to the grounding line the higher the requirements are on the resolution of the topography and the friction coefficient to obtain accurate solutions of $\mathbf{u}$ and $h$ there.

A weight is local if its extension in space is close to the observation point. The weights on $\delta C$ at the ice base are local for the steady state and time dependent FS model. They are also local for the time dependent SSA model and the transfer from $\delta b$ to $\delta u$ and $\delta h$ in the steady state. The sensitivity of $\delta u$ and $\delta h$ in the steady state of SSA depends on $\delta C$ from a larger domain. It is difficult to observe a perturbation $\delta C$ with a short wavelength on $u$ and $h$. In the example in Fig. 3, a spatial perturbation wavelength $\lambda = 2 \times 10^4$ m (about $10H$) in $C$ is damped by 0.2 in $\delta h$ and 0.02 in $\delta u$ compared to a wavelength $\lambda > 10^5$ where
there is no damping due to $\lambda$.

The perturbations in $u$ and $h$ in the steady state of the SSA model consists of a direct effect from $\delta b$ at the observation point, and a non-local effect of $\delta b$ and $\delta C$ in Figures 6 and 7. It follows from analytical solution in Eq. (A3) that we cannot distinguish between the non-local contributions of $\delta b$ and $\delta C$ in the integral to $\delta \mathbf{u}$. The same conclusion about the non-local perturbations holds for $\delta h$ in Eq. (A5).

The transfer matrices from $\delta b$ and $\delta C$ to $\delta u$ and $\delta h$ are examined by the singular value decomposition. If the quotient between the largest and the smallest singular values of the matrix is large then it is ill-conditioned and if it is small (but $\geq 1$) then the problem is well-conditioned. In an ill-conditioned problem, some perturbations at the base will be barely visible at the surface and a small perturbation at the top may correspond to a large perturbation at the bottom. In a well-conditioned problem, all perturbations at the base have a measurable effect at the surface. The ranking of the conditioning of the transfers in Fig. 9 from
the best to the worst is

1. $\delta b \rightarrow \delta h$, 2. $\delta b \rightarrow \delta u$, 3. $\delta C \rightarrow \delta h$, 4. $\delta C \rightarrow \delta u$.

In the past, the coupling between $\delta \mathbf{u}$ and $\delta C$ is most frequently used for inference of $C$ from velocity data but height data could improve the robustness of the inference.

*Code availability.* The FS equations are solved using Elmer/Ice Version: 8.4 (Rev: f6bfdc9) with the scripts at https://github.com/enigne/
FS_Adjoint. The forward and adjoint SSA solvers are implemented in MATLAB. The code is available at https://github.com/enigne/SSA_Adjoint.





## Appendix A: Some equations

Detailed derivations of the formulas are found in Cheng and Lötstedt (2019). A variable with index $*$ is evaluated at $x_*$.

### A1 The forward steady state SSA solution

The analytical steady state solution to the forward Eq. (17) without considering the viscosity terms is

$$
\begin{aligned}
H(x) &= \left( H_{GL}^{m+2} + \frac{m+2}{m+1} \frac{Ca^m}{\rho g} (x_{GL}^{m+1} - x^{m+1}) \right)^{\frac{1}{m+2}}, \ 0 \le x \le x_{GL}, \\
H(x) &= H_{GL}, \ x_{GL} < x < L, \\
u(x) &= \frac{ax}{H}, \ 0 \le x \le x_{GL}, \quad u(x) = \frac{ax}{H_{GL}}, \ x_{GL} < x < L,
\end{aligned}
\tag{A1}
$$

where $H_{GL}$ is the thickness of the ice at the grounding line $x_{GL}$.

### A2 The adjoint steady state SSA solutions

The analytical steady state solutions of the SSA adjoint Eq. (18) with observation of $u$ at $x_*$ is

$$
\begin{aligned}
(x) &= \frac{Ca^m x_*}{\rho g H_*^{m+3}} (x_{GL}^m - x^m), \ x_* < x \le x_{GL}, \\
(x) &= -\frac{1}{H_*} + \frac{Ca^m x_*}{\rho g H_*^{m+3}} (x_{GL}^m - x_*^m), \ 0 \le x < x_*, \\
v(x) &= \frac{ax_*}{\rho g H_*^{m+3}} H^m, \ x_* < x \le x_{GL}, \\
v(x) &= 0, \ 0 \le x < x_*,
\end{aligned}
\tag{A2}
$$

where $H_*$ is the thickness of the ice at $x_*$. The corresponding perturbation $\delta u_*$ in Eq. (20) has the weights for $\delta C$ and $\delta b$ as

$$
\begin{aligned}
\delta u_* &= \int_0^{x_{GL}} (\psi_x u + v_x \eta u_x + v \rho g h_x) \delta b - v u^m \, \delta C \, \mathrm{d}x \\
&= \frac{u_*}{H_*} \delta b_* - \frac{u_*}{H_*} \int_{x_*}^{x_{GL}} \frac{C(ax)^m}{\rho g H_*^{m+1}} \left( (m+1)\frac{\delta b}{H} + \frac{\delta C}{C} \right) \mathrm{d}x,
\end{aligned}
\tag{A3}
$$

If $h$ is observed at $x_*$, then

$$
\begin{aligned}
(x) &= -\frac{Ca^{m-1}}{\rho g H_*^{m+1}} (x_{GL}^m - x^m), \ x_* < x \le x_{GL}, \\
(x) &= -\frac{Ca^{m-1}}{\rho g H_*^{m+1}} (x_{GL}^m - x_*^m) - \frac{\delta(x - x_*)\eta_*}{n \rho g H_*}, \ 0 \le x \le x_*, \\
v(x) &= -\frac{H^m}{\rho g H_*^{m+1}}, \ x_* < x \le x_{GL}, \\
v(x) &= 0, \ 0 \le x < x_*.
\end{aligned}
\tag{A4}
$$

The weights for $\delta C$ and $\delta b$ in Eq. (19) for the perturbation on $h_*$ is

$$
\delta h_* = \frac{\eta_*}{n \rho g H_*} (u \delta b)_x (x_*) + \int_{x_*}^{x_{GL}} \frac{C(ax)^m}{\rho g H_*^{m+1}} \left( (m+1)\frac{\delta b}{H} + \frac{\delta C}{C} \right) \mathrm{d}x,
\tag{A5}
$$



*Author contributions.* GC contributed most of the computations and GC and PL contributed equally to the theory and the writing of the paper.

*Competing interests.* The authors declare that they have no conflict of interest.

*Acknowledgements.* This work has been supported by Nina Kirchner's Formas grant 2017-00665 and the Swedish e-Science initiative eSSENCE. Thomas Zwinger has been helpful with the adjoint FS solver in Elmer/Ice. Comments by Lina von Sydow have helped us improve a draft of the paper.



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
