# Peer review of "Parameter sensitivity analysis of dynamic ice sheet models-Numerical computations"

_The Cryosphere, 2019_

## Referee Comment (RC1) · Anonymous Referee #1 · 26 Nov 2019

The authors consider a very important and interesting problem, ie the (inverse) problem of estimating the sensitivity of basal flow parameters to surface date. In fact, this is such an interesting question that it has been addressed many times in many publications in glaciology before. I have a positive view of this work. However, I think the best approach forward is to ask the authors to rework their manuscript and provide much better context and comparison of their work with previous work. Below I give some references to papers that the authors might find useful in this respect.

The formulation of the adjoint equations for the time depended SSA case is, I believe, done here for the first time. I found it next to impossible to follow the derivations in the paper. However, reading (Cheng & Lötstedt, 2019) this all became much easier to understand. I wonder if it might not be a good idea to focus the paper more on the relevant message to the glaciological community and either offload some more of the technical details to appendixes or just refer to the arXiv manuscript.

I found it very nice how w\_{ub} and w\_{uC} are determined from the solutions of the adjoint problem \phi and v. This is actually a straightforward application of the adjoint method, but at least in glaciology I have not seen this done so often, although possibly (Martin & Monnier, 2014; Monnier & des Boscs, 2017) may have done this already. This is a clever way of estimating the sensitivity of, for example, velocities at one given location to any perturbation in C. (But are not a brackets missing in Eq. 20 and 21?). I suspect that this can easily be done in any modern ice-flow model by just modifying the cost function to include surface data from only one location at a time.

I found the transfer matrix approach also to be very interesting. As this approach has been used before by (Gudmundsson, 2008; Gudmundsson & Raymond, 2008; Pralong & Gudmundsson, 2011; Thorsteinsson et al., 2004) it would have been valuable for the reader to be able to understand to what the differences are with respect to those previously published studies. Since the authors mostly consider the case m=1 they can compare this with previously published analytical transfer functions (note that the m>1 solutions by (Gudmundsson, 2008) contain an error, but the m=1 case is OK). It appears that the main differences are that this study is numerical and the sensitivity matrices Q\_{ub} and W\_{uC} evaluated numerically. This gives great flexibility, but makes it more difficult to arrive at general conclusions. The work seems related to (Martin & Monnier, 2014) who also used a purely numerical approach.

The authors state that previously 'The time dependent height equation for the moving upper surface is not included in the inversion.' While this may be true for some inverse models, there are a number of publications that use ds/dt (s being the upper surface) information in the inversion. This has been done for example by using the kinematic boundary condition at the upper surface or the vertically integrated mass conservation equations. To my knowledge, all the modern ice-flow models (i.e. ISSM, Wavy, Úa, BISICLES, Elmer/Ice) allow for this option. The dh/dt (h ice thickness) is, for example, used to determine ice thickness in BISICLES and ISSM and when solving for basal slipperiness and ice rheology parameters in Wavy and Úa. See for example (Kyrke-Smith et al., 2018; Monnier & des Boscs, 2017). However, the authors are I think right in stating that the adjoint equations have not been derived for the transient SSA equation before. However, I believe that in effect Dan Golberg has done so previously using automated differentiation (Goldberg et al., 2015).

I must confess that I found most of the conclusions and the points addressed in the discussions rather weak. It is always going to be difficult to make any general statements about parameter

sensitivity using a numerical approach. I think the approach the authors use is excellent if looking at some specific domains and for some specific model studies. I could for example imagine this to be a useful exercise when looking at particular parts of, for example, the West Antarctic Ice Sheet.

A side issue that I have with the general approach is that an inverse problem never explicitly defined. Often in inverse theory one states that the objective is, for example, to evaluate to conditional probability P(C|u). This then allows one to define all kinds of clearly defined properties such as the number of resolved model parameters as function of the number of measurements and measurements errors, etc. etc. I understand that the authors are here only interested in parameter sensitivity, but this somewhat narrow viewpoint of an inverse problem makes the findings arguably less interesting.

Overall, I have a very positive view of this work. It is highly competent and I enjoyed reading the paper. I would suggest making more references and links to existing work. Also, consider taking some of the technical aspect and put them into appendixes. Especially since the computations cannot really be understood without reading authors previous paper on this subject.

- Cheng, G., & Lötstedt, P. (2019). Parameter sensitivity analysis of dynamic ice sheet models. *ArXiv*, 1–28. Retrieved from http://arxiv.org/abs/1906.08197
- Goldberg, D. N., Heimbach, P., Joughin, I., & Smith, B. (2015). Committed retreat of Smith, Pope, and Kohler Glaciers over the next 30 years inferred by transient model calibration. *Cryosphere*, *9*(6), 2429–2446. https://doi.org/10.5194/tc-9-2429-2015
- Gudmundsson, G. H. (2008). Analytical solutions for the surface response to small amplitude perturbations in boundary data in the shallow-ice-stream approximation. *The Cryosphere*, *2*(2), 77–93. https://doi.org/10.5194/tc-2-77-2008
- Kyrke-Smith, T. M., Gudmundsson, G. H., & Farrell, P. E. (2018). Relevance of Detail in Basal Topography for Basal Slipperiness Inversions: A Case Study on Pine Island Glacier, Antarctica. *Frontiers in Earth Science*, 6. https://doi.org/10.3389/feart.2018.00033
- Martin, N., & Monnier, J. (2014). Adjoint accuracy for the full Stokes ice flow model: limits to the transmission of basal friction variability to the surface. *The Cryosphere*, 8(2), 721–741. https://doi.org/10.5194/tc-8-721-2014
- Monnier, J., & des Boscs, P.-E. (2017). Inference of the bottom properties in shallow ice approximation models. *Inverse Problems*, *33*(11), 115001. https://doi.org/10.1088/1361-6420/aa7b92

---

## Referee Comment (RC2) · Alexander Robinson (Referee) · 4 Dec 2019

I find this manuscript very relevant to TC. It presents interesting results from a new inversion comparing Full Stokes and SSA models. Nonetheless, I believe the manuscript requires significant revision before publication.

Motivation: The manuscript does not give the reader a sense of why this study is necessary or important. This is particularly true for the Introduction, where it should be made clear why this exercise is undertaken and what can be expected as an advance compared to previous work.

Language: There are not so many grammatical mistakes, but nonetheless, phrasing choices used her make it somewhat difficult to follow the text. I suggest the authors

ask an independent reviewer to help improve this aspect of the manuscript.

Conclusions (and Discussion): I would like to see somewhat more general extractions from the work to make it more broadly applicable. How would this method work on a more realistic domain? What would it take to make the approach work for real data? Do you believe the last conclusion (ranking of the conditioning of transfers) is general for any problem?

Specific comments:

Abstract: Please add a sentence or two specifically reporting the important results found from this exercise.

Paragraph starting P1L20: This review needs revising. Aside from the references, as mentioned by reviewer 1, this paragraph does not leave the reader with a clear idea of which study addressed which issue, and why. Please carefully improving phrasing.
* * *

---

## Author Comment (AC1) · 27 Dec 2019

article [english]babel amsmath amssymb color

[Figure]

**Response to Anonymous Referee #1**

December 27, 2019

The authors consider a very important and interesting problem, ie the (inverse) problem of estimating the sensitivity of basal flow parameters to surface date. In fact, this is such an interesting question that it has been addressed many times in many publications in glaciology before. I have a positive view of this work. However, I think the best approach forward is to ask the authors to rework their manuscript and provide much better context and comparison of their work with previous work. Below I give some references to papers that the authors might find useful in this respect.

**Response**: The new references suggested by the reviewer have been added and more comparisons are made with earlier work in the Introduction and at other places in the paper.

The formulation of the adjoint equations for the time depended SSA case is, I believe, done here for the first time. I found it next to impossible to follow the derivations in the paper. However, reading (Cheng & Lötstedt, 2019) this all became much easier to understand. I wonder if it might not be a good idea to focus the paper more on the relevant message to the glaciological community and either offload some more of the technical details to appendixes or just refer to the arXiv manuscript.

**Response**: More discussion and conclusions are found in the final two sections. The

description of the SVD is moved to a subsection in the Appendix.

I found it very nice how $w_{ub}$ and $w_{uC}$ are determined from the solutions of the adjoint problem $\phi$ and v. This is actually a straightforward application of the adjoint method, but at least in glaciology I have not seen this done so often, although possibly (Martin & Monnier, 2014; Monnier & des Boscs, 2017) may have done this already. This is a clever way of estimating the sensitivity of, for example, velocities at one given location to any perturbation in C. (But are not a brackets missing in Eq. 20 and 21?). I suspect that this can easily be done in any modern ice-flow model by just modifying the cost function to include surface data from only one location at a time.

**Response**: Yes, the weights can be computed for any ice model not just FS and SSA. We discuss the mentioned papers and do not think that Monnier et al did it in this way. The convention is sometimes that the brackets are not needed around a sum in the integral.

I found the transfer matrix approach also to be very interesting. As this approach has been used before by (Gudmundsson, 2008; Gudmundsson & Raymond, 2008; Pralong & Gudmundsson, 2011; Thorsteinsson et al., 2004) it would have been valuable for the reader to be able to understand to what the differences are with respect to those previously published studies. Since the authors mostly consider the case $m = 1$ they can compare this with previously published analytical transfer functions (note that the $m > 1$ solutions by (Gudmundsson, 2008) contain an error, but the $m = 1$ case is OK). It appears that the main differences are that this study is numerical and the sensitivity matrices $Q_{ub}$ and $W_{uC}$ evaluated numerically. This gives great flexibility, but makes it more difficult to arrive at general conclusions. The work seems related to (Martin & Monnier, 2014) who also used a purely numerical approach.

**Response**: The transfer matrices are evaluated both numerically and analytically for SSA. We have written a paragraph in the Result section about the interpretation of the formulas in the Appendix.

The authors state that previously 'The time dependent height equation for the moving upper surface is not included in the inversion.' While this may be true for some inverse models, there are a number of publications that use $ds/dt$ (s being the upper surface) information in the inversion. This has been done for example by using the kinematic boundary condition at the upper surface or the vertically integrated mass conservation equations. To my knowledge, all the modern ice-flow models (i.e. ISSM, Wavy, Ua, BISICLES, Elmer/Ice) allow for this option. The $dh/dt$ ($h$ ice thickness) is, for example, used to determine ice thickness in BISICLES and ISSM and when solving for basal slipperiness and ice rheology parameters in Wavy and Ua. See for example (Kyrke-Smith et al., 2018; Monnier & des Boscs, 2017). However, the authors are I think right in stating that the adjoint equations have not been derived for the transient SSA equation before. However, I believe that in effect Dan Golberg has done so previously using automated differentiation (Goldberg et al., 2015).

**Response**: Goldberg's work is discussed. It uses autometic differentiation, numerical approximation in time, and transient data but the analytical adjoint is not derived. The analytical adjoint equations allow us to draw conclusions about the solutions for FS and SSA (e.g. in the Appendix) and the sensitivity in the more theoretical paper in arXiv by the authors and in a new paragraph in Section 3.2. Another conclusion is that the adjoint height equation and its solution is important for height perturbations but not for velocity perturbations (see e.g. Conclusions). Yes, time dependent data are permitted in modern codes but without including the time derivative of the height in the differential equation. For instance, in Monnier and des Boscs for the extended SIA model $dh/dt$ is subtracted from the surface mass balance and a stationary problem is solved (7).

I must confess that I found most of the conclusions and the points addressed in the discussions rather weak. It is always going to be difficult to make any general statements about parameter sensitivity using a numerical approach. I think the approach the authors use is excellent if looking at some specific domains and for some specific model studies. I could for example imagine this to be a useful exercise when looking at

particular parts of, for example, the West Antarctic Ice Sheet.

**Response**: Numerical evaluations of the transfer matrices are the only possible option for complicated geometries and nonlinear equations. Under certain assumptions, the analytical expressions for SSA in the Appendix tell how the sensitivity varies with $u, h, C,$ and $b$. More discussion and conclusions have been included in the new version of the paper regarding this issue.

A side issue that I have with the general approach is that an inverse problem never explicitly defined. Often in inverse theory one states that the objective is, for example, to evaluate to conditional probability $P(C|u)$. This then allows one to define all kinds of clearly defined properties such as the number of resolved model parameters as function of the number of measurements and measurements errors, etc. etc. I understand that the authors are here only interested in parameter sensitivity, but this somewhat narrow viewpoint of an inverse problem makes the findings arguably less interesting.

**Response**: The relation to the inverse problem is discussed in the new Section 2.2.5 now. There is also a discussion of this matter in the arXiv paper.

Overall, I have a very positive view of this work. It is highly competent and I enjoyed reading the paper. I would suggest making more references and links to existing work. Also, consider taking some of the technical aspect and put them into appendixes. Especially since the computations cannot really be understood without reading authors previous paper on this subject.

**Response**: Thank you for the review. More references have been added and discussed and the SVD account is now in the Appendix.

---

## Author Comment (AC2) · 27 Dec 2019

article [english]babel amsmath amssymb color

[Figure]

**Response to Referee #2**

December 27, 2019

I find this manuscript very relevant to TC. It presents interesting results from a new inversion comparing Full Stokes and SSA models. Nonetheless, I believe the manuscript requires significant revision before publication.

Motivation: The manuscript does not give the reader a sense of why this study is necessary or important. This is particularly true for the Introduction, where it should be made clear why this exercise is undertaken and what can be expected as an advance compared to previous work.

**Response**: The second paragraph in Introduction is now a motivation.

Language: There are not so many grammatical mistakes, but nonetheless, phrasing choices used her make it somewhat difficult to follow the text. I suggest the authors ask an independent reviewer to help improve this aspect of the manuscript.

**Response**: The language has been revised here and there.

Conclusions (and Discussion): I would like to see somewhat more general extractions from the work to make it more broadly applicable. How would this method work on a more realistic domain? What would it take to make the approach work for real data? Do you believe the last conclusion (ranking of the conditioning of transfers) is general

for any problem?

**Response**: The Discussion and the Conclusions have been extended with more results from the numerical experiments and the analytical solutions. The method is general and would work in 3D with some simplifications based on ideas in adaptive mesh refinement. Real data for $u$ and $h$ may be used for inversion to find $C$ and $b$ at the base. The relation between the inverse problem and the sensitivity is mentioned now in Section 2.2.5.

**Specific comments**

1. Abstract: Please add a sentence or two specifically reporting the important results found from this exercise.

   **Response**: New sentences are now in the Abstract summarizing some of the conclusions

2. Paragraph starting P1L20: This review needs revising. Aside from the references, as mentioned by reviewer 1, this paragraph does not leave the reader with a clear idea of which study addressed which issue, and why. Please carefully improving phrasing.

   **Response**: Parts of the Introduction have been rewritten with more references and a description of the work there.

---

## Author Comment (AC3) · 27 Dec 2019

The comment was uploaded in the form of a supplement:
https://www.the-cryosphere-discuss.net/tc-2019-151/tc-2019-151-AC3-supplement.pdf